# Centralized softening as a solution to chloride pollution: An empirical analysis based on Minnesota cities

**Baishali Bakshi** [1]*, **Elise M. Doucette**[2], **Scott J. Kyser**[2]

1 Water Quality Standards Unit, Water Assessment Section, Environmental Analysis and Outcomes Division, Minnesota Pollution Control Agency, Saint Paul, Minnesota, United States of America, 2 Effluent Limits Unit, Water Assessment Section, Environmental Analysis and Outcomes Division, Minnesota Pollution Control Agency, Saint Paul, Minnesota, United States of America

* Baishali.Bakshi@state.mn.us

## Abstract

Chloride is a key component of salt, used in many activities such as alkali production, water treatment, and de-icing. Chloride entering surface and groundwater is a concern due to its toxicity to aquatic life and potential to degrade drinking water sources. Minnesota being a hard-water state, has a high demand for water softening. Recent research has found that home-based water softeners contribute significantly to chloride loading at municipal waste-water treatment plants (WWTPs). Because of this, many WWTPs would now require water quality based effluent limits (WQBELs) to comply with the state's chloride water quality standards (WQS), unless they install chloride treatment technologies, which are limited and cost-prohibitive to most communities. A potential solution to this problem, is shifting from home-based water softening to a system where water is softened at drinking water plants, before reaching homes, i.e. centralized softening, analyzed in this paper based on its ability to address both chloride pollution and water softening needs, at reasonable cost. We estimate lifetime costs of three alternative solutions: centralized softening, home-based softening, and a Business as Usual (BAU) or baseline alternative, using annualized 20-year loan payments and Net Present Value (NPV), applied to 84 Minnesota cities with matching data on drinking water plants and WWTPs. We find that centralized softening using either Reverse Osmosis (RO) or lime-softening technologies is the more cost-effective solution, compared to the alternative of home-based softening with end-of-pipe chloride treatment, with a cost ratio in the range 1:3–1:4. Between the two centralized softening options, we find RO-softening to be the lower cost option, only slightly more costly (1.1 cost ratio) than the BAU option. Considering additional environmental and public health benefits, and cost savings associated with removal of home-based softeners, our results provide helpful information to multiple stakeholders interested in an effective solution to chloride pollution.

**Data Availability Statement:** All relevant data are within the paper and its Supporting Information files.

**Funding:** The authors received no specific funding for this work.

**Competing interests:** The authors have declared that no competing interests exist.

## Introduction

Chloride is a key component of salt, used in many activities such as chlor-alkali [1] production, de-icing, fertilizer use, and water softening [2]. Chloride enters groundwater and surface water through several pathways affecting: (1) aquatic life–through toxicity [3–5]; (2) public health–through higher salt content and corrosivity of drinking water, affecting people and infants who have higher salt sensitivity and problems metabolizing salt, and potentially causing metal leaching from taps and fixtures [6, 7]; and (3) the life of septic systems–through corrosion.

Hard water is water containing a high concentration of calcium and magnesium ions, i.e. high mineral content [8]. Hardness is measured as milligram per liter (mg/L) or grains per gallon (gpg) as calcium carbonate, $CaCO_3$. According to the United States Geological Survey, water with more than 120 mg/L as $CaCO_3$ is hard [9], which translates to about 7.01 gpg [10]. Based on hardness prevalence [9], home-based water softening with **point-of-entry ion-exchange** water softeners is common in many states, to treat water for domestic uses such as washing, bathing, and for use in household appliances [11, 12]. Point-of-entry softeners treat water for an entire home or building and are installed directly at the point the main water service line *enters a home* [13, 14]. Ion-exchange is the most common type of point-of-entry softener: these work by running a sodium chloride (i.e. salt brine) solution through a resin base, a process that *exchanges calcium and magnesium ions that contribute to water hardness*, *with soluble sodium ions*, yielding soft water, and a waste brine containing the displaced and/or excess chloride ions along with the hardness ions [15].

However, home-based ion-exchange water softeners cause significant chloride loading to municipal wastewater treatment plants (WWTPs), which eventually discharge into surface waters [11, 16, 17]. Removing chloride from WWTP discharges is infeasible due to high cost and limited availability of treatment technologies [18], and once this wastewater passes into waterbodies, removal of chloride is likely to be impossible [19]. High enough levels of chloride in freshwater lakes and streams can be toxic to aquatic life–fish, shellfish, freshwater mussels, and insects like mayflies.

The federal Clean Water Act is the primary law that regulates the amount of chloride municipal WWTPs can discharge. It requires states to designate **beneficial uses** for all waters and develop water quality standards (WQS) to protect each use [20, 21]. WQS comprise several parts, including beneficial uses, which 'Identify how people, aquatic communities, and wildlife use our waters' [22]. The Environmental Protection Agency (EPA) has a secondary drinking water standard for chloride, known as a secondary maximum contaminant level (SMCL), of 250 mg/L, which is not enforceable but set to prevent drinking water from having an unpleasantly salty taste [23], and has established a recommended national chronic chloride standard of 230 mg/L and an acute standard of 860 mg/L to protect freshwater aquatic life from the impacts of chloride [24], which several states have adopted. For example, Minnesota is among thirty-five states that have surface water quality criteria for chloride that either follow EPA's national criteria, localized updates to the national criteria or have criteria to protect irrigation or public water supplies.

Chloride concentrations are increasing in waterbodies across the United States (see Fig 1 [25]). De-icing salts are the dominant source of chloride, especially in cold-weather states like Minnesota and Wisconsin, but home water softening is also a significant contributor to excess chloride in these states as well as in several others such as Arizona, California, Connecticut, Illinois, Iowa, Michigan, Ohio, and Texas. There are no studies quantifying chloride loading from water softening at regional or national scales, but based on general observation, areas of the country with hard water are very likely to also have a high chloride load in municipal wastewater discharges due to widespread use of ion-exchange water softeners. Unlike chloride loading

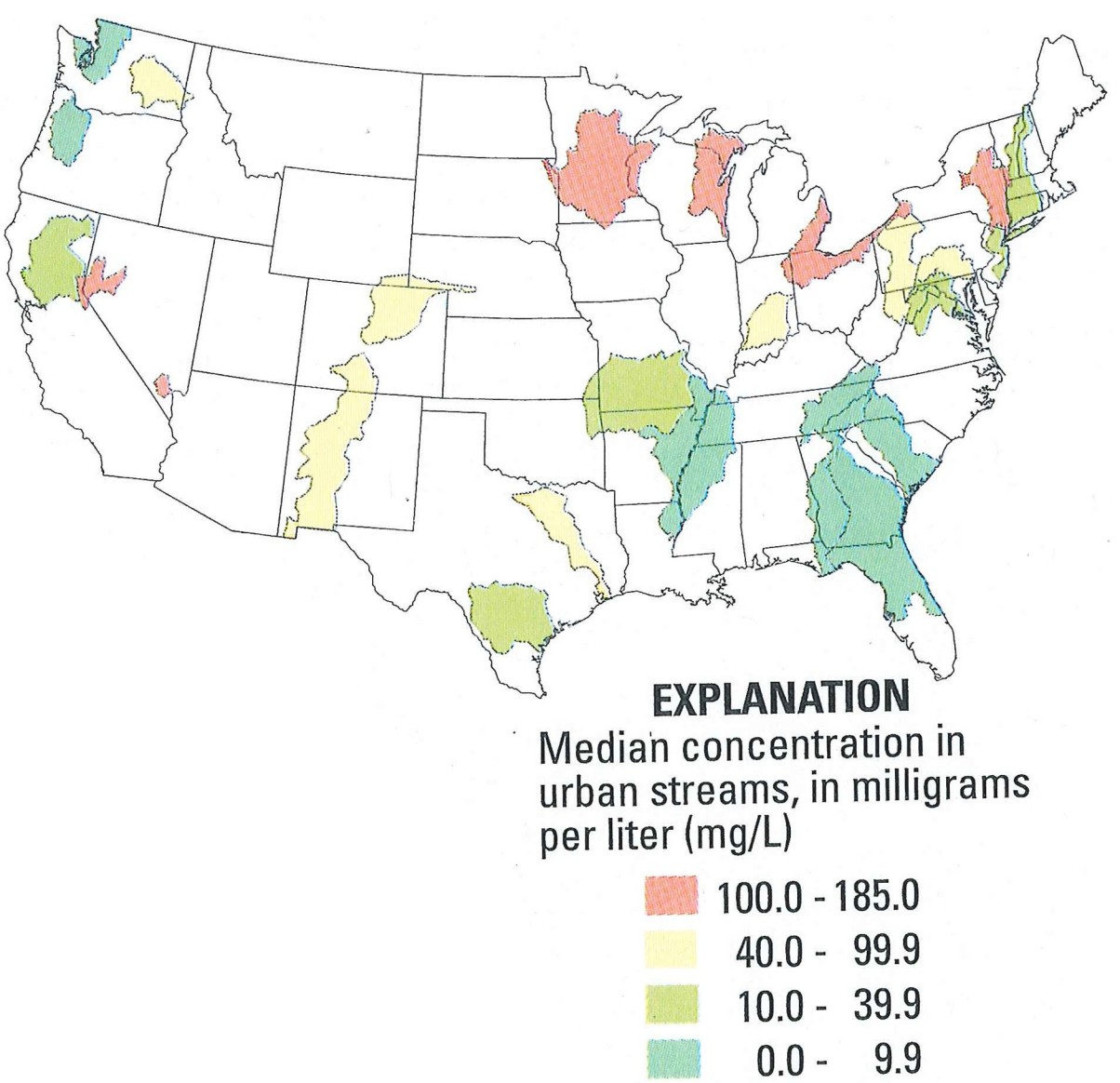

**Fig 1. Chloride concentrations in urban streams in the US.** *Source*: *USGS Water Quality in the Upper Mississippi Basin 1995–1998*. Reprinted from the original map image of Fig 1 under public domain privilege, as certified by statement from Sabra Lopez, Science Information Services, United States Geological Service, 2020.

from winter de-icing, chloride is continuously discharged from municipal WWTPs and can cause harmful in-stream chloride concentrations during the non-winter months when drought and low flows are most likely to occur. Moreover, EPA's nationally recommended 230 mg/L chloride water quality criteria for surface freshwater dates back to 1988 [24], and recent advances in chloride toxicology finds this criteria needs updating to be protective of sensitive aquatic species such as mayflies [26, 27] and freshwater mussels [28, 29]. Excessive chloride pollution of surface and groundwater could also lead to increased corrosion of water infrastructure such as drinking water distribution systems [30]; controlling corrosion is an important part of avoiding public health crises related to unsafe drinking water such as occurred in Washington DC [31] and Flint Michigan [32]. The current increasing trend of chloride in surface water,

including groundwater in several cities, such as Madison, Wisconsin [33], and Chicago, Illinois [34], highlights the needs for a solution that avoids adding chloride to the environment and helps prevent such public and environmental health threats nationwide.

If any state, tribe or territory were to update their aquatic life criteria using the best available science, it is likely that wastewater dischargers would face more restrictive chloride discharge requirements and many dischargers would need to find ways to further reduce chloride in their discharge. In areas of hard water, municipal WWTPs are increasingly being required to reduce chloride loading from water softeners to comply with chloride wastewater discharge requirements. Since there is no national strategy on reducing chloride from water softeners, affected cities have employed a patchwork of approaches ranging from state regulations allowing local water softener bans (California [35], Connecticut [36], Texas [37]) to providing local incentives to install high efficiency softeners (Scottsdale, Arizona [38]; Lake Geneva, Wisconsin [39]) to requiring cities to install centralized softening (Fort Dodge, Iowa [40]) to installing chloride treatment technologies at the WWTP (Santa Clarita, California [41]).

Chloride treatment technologies at the WWTP, i.e. end-of-pipe, are limited, and are extremely expensive to the extent of being cost-prohibitive for most communities [18]. This leads to the problem of achieving sufficient reductions in WWTP chloride discharges to comply with state WQS, while providing an effective way of softening water at reasonable cost. A potential solution to this problem, is shifting from home-based water softening to a system where water is softened at drinking water plants, before reaching homes, i.e. centralized softening (CS). An economic analysis of the lifetime costs of centralized softening compared to alternative solutions could provide useful information to states faced with this problem. We attempted to quantify chloride loading from water softeners on a regional or national level, but abandoned the effort because it is exceedingly difficult and resulted in unreasonable uncertainty.

As Minnesota is primarily a hard water state (see Fig 2) with high prevalence of ion-exchange home water softeners [16], and an increasing trend in chloride concentrations (see Fig 1), a similar analysis for Minnesota could also represent other hard water states facing this problem. With regard to state WQS, drinking water and aquatic life are both important beneficial uses for Minnesota's waterbodies. Minnesota has adopted the federal surface WQS of 230 mg/L (chronic) and 860 mg/L (acute) to protect aquatic life. The federal SMCL of 250 mg/L for drinking water is also recognized to prevent salty taste. Therefore, in this paper, we used Minnesota as a case study to evaluate the lifetime cost effectiveness of centralized softening compared to alternative solutions including end-of-pipe, and a status-quo or Business-as-Usual (BAU) option of home-softener usage with no strategy for chloride reduction. Our paper investigates Minnesota's chloride pollution problem using data from Minnesota communities and finds centralized softening to be the most environmentally-effective and cost-effective solution.

About 75% of Minnesota communities receive their drinking water from groundwater [42]. Chloride levels in groundwater and surface water are naturally very low. De-icing salts are the dominant source of chloride in Minnesota, but home-based ion-exchange water softeners are documented to be a major contributor, causing significant chloride loading to municipal WWTPs statewide, which eventually discharge into surface waters [16, 17]. As noted before, removing chloride in WWTP discharges is infeasible owing to high cost and limited availability of treatment technologies [18] and is almost impossible once the wastewater passes into waterbodies.

Chloride concentrations in Minnesota waters have been increasing over time [43], particularly in urban areas; as noted in Fig 1, Minnesota has relatively high chloride concentrations in urban streams [25]. Currently 27% of groundwater monitoring wells in the Twin Cities Metropolitan Area (TCMA) exceed the SMCL. The TCMA also has 39 out of the 50 surface waters

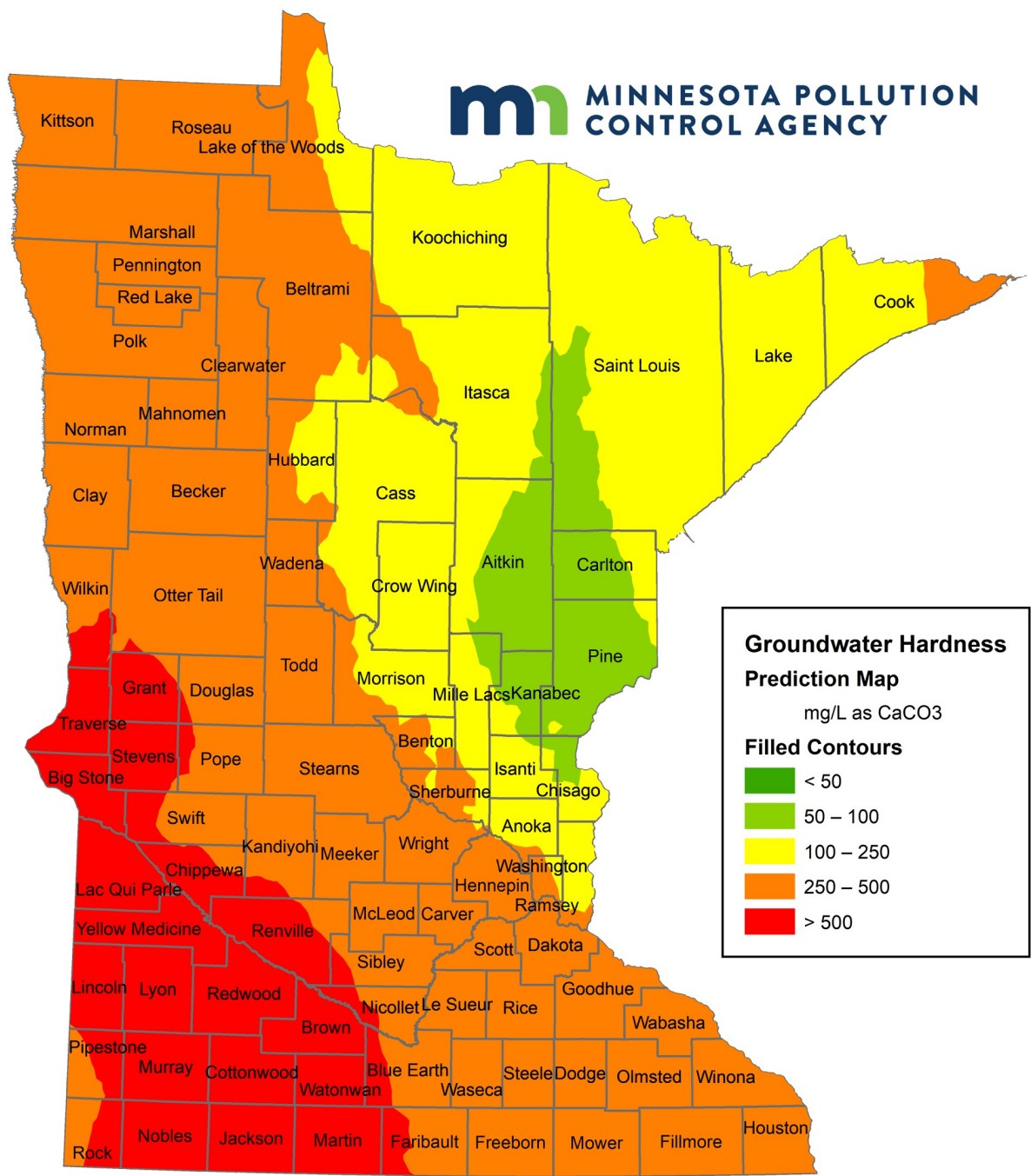

**Fig 2. Water hardness in Minnesota communities.** Reprinted from the original map image of Fig 1 under a CC BY license, with permission from Steven Weiss, Supervisor-Effluent Limits Unit, Minnesota Pollution Control Agency, original copyright 2020.

impaired statewide for chloride, with another 38 designated as being at high risk of impairment due to amounts of chloride that are approaching the WQS [44]. Statewide, based on limited monitoring, there are a total of 75 waterbodies at high risk of becoming impaired [45] (Fig 3).

Over one hundred municipal WWTPs, 20% of those in the state and serving approximately 10% of the state's population, are currently likely to need more stringent chloride discharge

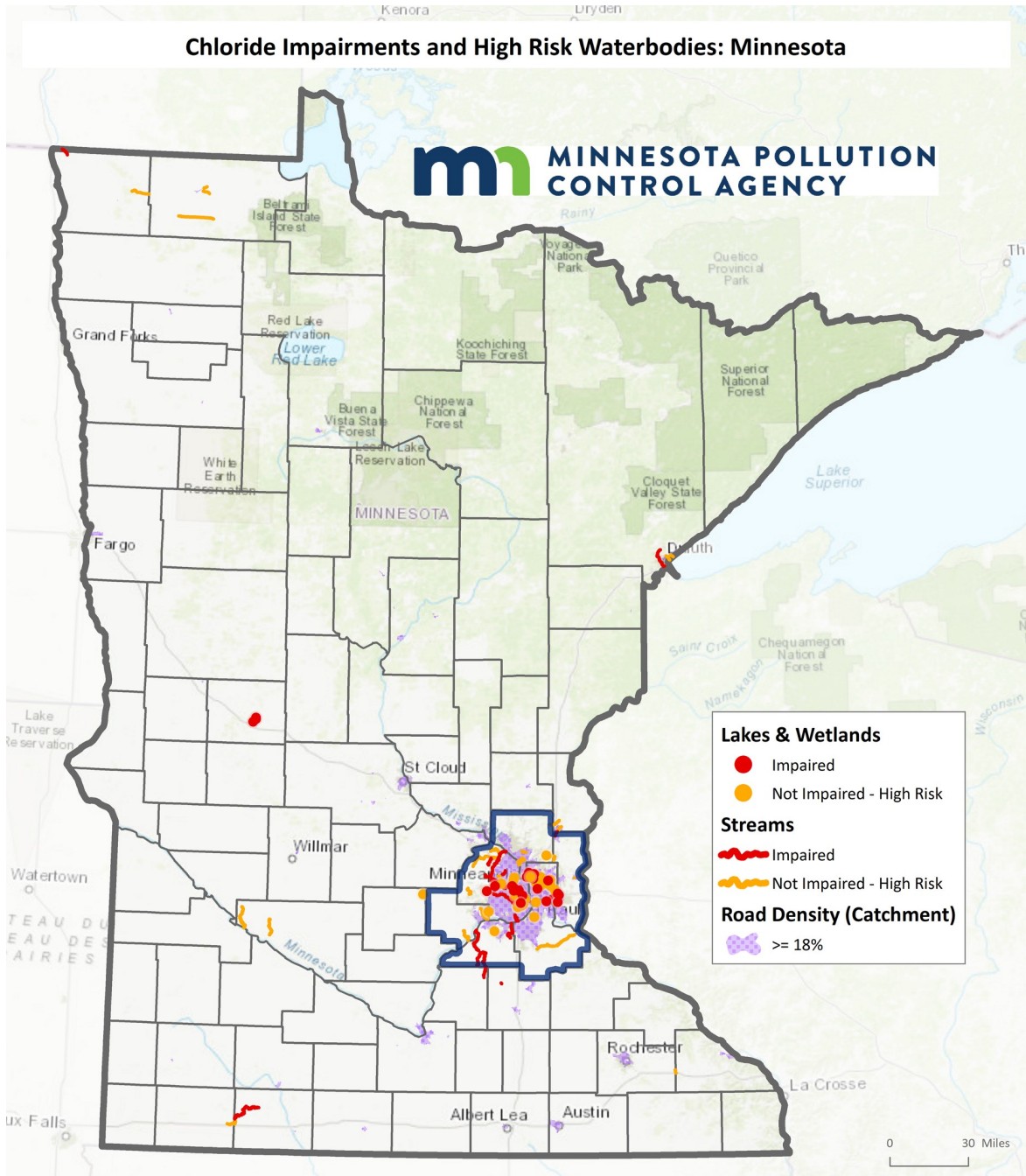

**Fig 3. Chloride impairments in Minnesota, 2019.** Reprinted from the original map image of Fig 3 under a CC BY license, with permission from Steven Weiss, Supervisor-Effluent Limits Unit, Minnesota Pollution Control Agency, original copyright 2020.

(effluent) limits. This means they have RP–a **reasonable potential** to cause or contribute to a violation of the chloride WQS. Both surface water chloride impairments and WWTP RPs are dependent on monitoring data and therefore could underestimate actual figures. Facilities with RP for chloride will need to take actions to mitigate high chloride discharges to surface water–either by fully meeting their chloride limits or applying for a variance, (a temporary

change in water quality standards to allow time for compliance) and completing chloride minimization plans.

It will be difficult for municipalities to meet these limits without managing the use of home-based water softeners [18], unless they install additional end-of-pipe chloride treatment technologies such as desalination or reverse osmosis using evaporation and crystallization (RO-EC). Such technologies are limited in availability and expensive, and particularly unaffordable for small communities. Affordability is typically measured by a threshold of 2% of community median household income (MHI), set by the EPA in its economic guidance for the process of setting water quality standards, based on a variety of factors including wastewater treatment costs [46].

Therefore, an evaluation of a potential solution–cities shifting to centralized water softening at their drinking water plants–could provide useful information to homeowners, drinking water plants, WWTPs, and local and state decision-makers. Centralized softening could provide softened water to households and businesses, allowing individual sites to remove their ion-exchange softeners. Centralized softening has the potential to be a lower-cost option to help WWTPs reduce chloride in their effluent, enabling them to meet their chloride WQBELs, and contribute to the protection of Minnesota's water resources and public health by preventing the release of excess chloride to waterbodies [47].

Centralized softening at drinking water plants typically comprises one of three options: **CS using ion-exchange: CS-ion-exchange, CS using reverse osmosis (RO) filtration: CS-RO, and CS using lime to soften water: CS-Lime**. All three technologies are capable of softening water to levels appropriate for daily use. The levels to which CS can soften water depend on multiple factors including existing water infrastructure, treatment technology applied, and public health guidelines for the distribution system in a given community [18]. We discuss treatment technologies and their costs in more detail in Section 3.

The application of centralized softening can reduce the overall costs of both water softening (compared to softening at the household level) as well as technology costs of chloride treatment (compared to treatment at the WWTP level) owing to the higher efficiency of the process and possible economies of scale in utilizing infrastructure and equipment for entire communities at once. In addition, current research finds centralized softening has the lowest footprint in a lifetime context compared to ion-exchange softening with home-softeners and mitigates climate change by 0.11 Million tonnes of $CO_2$ eq./year [48]. It can also be a winning alternative from the perspectives of both homeowners and WWTPs. Homeowners could avoid incurring continued home-based water softener costs and potentially higher user rates resulting from construction of end-of-pipe chloride treatment at the WWTPs, and WWTPs could meet their chloride limits without having to install these expensive chloride treatment technologies. Finally, by preventing chloride from entering wastewater, it would protect beneficial uses and minimize additional environmental impact.

In this paper, we illustrate the effectiveness of centralized softening as a potential solution that supports the ultimate goal of protecting water quality from chloride pollution. Protecting water quality from the impacts of chloride requires solving the key problem of ensuring WWTPs meet their chloride limits at reasonable cost. A critical barrier to solving this problem is the need for households to have softened water [49]. To see whether centralized softening could solve both of these problems at reasonable cost, we estimated the lifetime costs of three alternative solutions: centralized softening, home-based softening, and a BAU or baseline alternative, using annualized 20-year loan payments and Net Present Value (NPV), applied to 84 Minnesota cities with matching data on drinking water plants and WWTPs. We found that centralized softening using either Reverse Osmosis (RO) or lime-softening technologies is the more cost-effective solution, compared to the alternative of home-based softening with end-

of-pipe chloride treatment, with a cost ratio in the range 1:3–1:4. Considering additional environmental and public health benefits, and cost savings associated with removal of home-based softeners, our results show that centralized softening could be a critical component of effective chloride management compared to other alternatives, and provide helpful information to multiple stakeholders interested in an effective solution to chloride pollution. The findings of this study are applicable nationally because the costs of affordable chloride reduction alternatives for municipalities are generalizable, scalable and allow for chloride reductions everywhere.

## Chloride loading to waters and municipal WWTPs

Tackling the chloride problem–the need to ensure that water quality standards are met–requires understanding how chloride enters Minnesota's surface and groundwater. Chloride loading to waterbodies in Minnesota occurs from multiple sources, including road salt, industrial processes, and WWTP effluent. Nationally, road salt is a major source, particularly in urban areas [50, 51]. WWTP effluent is also recognized as a significant source [52], mainly owing to the use of salt for home-based water softening and consequent loading of salt to WWTPs.

While road salt is the largest source (42%) of chloride loading to Minnesota waterbodies, current research has documented WWTPs as a major source (22%) and the largest point source of chloride statewide (see Fig 13, page 24, in [16]). Owing to very high water hardness (Fig 1), particularly in groundwater that serves as drinking water for 75% of residents, Minnesota has a high prevalence (72% of residents) of home-based water softening. Thus, home-based and commercial water softening is the primary cause of WWTP chloride loading at 65% while industrial sources contribute 22% of the load (see Fig 12, page 23, in [16]).

Ion-exchange softeners are used residentially and at drinking water plants, as well as industrially and commercially. Residential ion-exchange softeners are the most significant source of chloride loading to WWTPs, followed by industrial ion-exchange softeners, as shown by the three example cities in Table 1. It is worth noting that the City of Long Prairie is an exception, where chloride loading from industrial ion-exchange is the largest source to WWTPs, mainly because of salt-intensive industries such as meatpacking, but these cities are outliers and do not affect the result that on average, residential ion-exchange is the largest source of chloride loading to WWTPs.

In addition to softener use, salt may be used in daily domestic activities such as food preparation. The contribution of these activities as depicted by the category "residential (non-ion-exchange)", is minuscule compared to "residential (ion-exchange)". In general, these other areas of residential use contribute even less than the existing chloride content of the source water when it enters a drinking water plant.

Fig 4, below, shows how the use of home-based softeners leads to chloride loading from homes and private wells to WWTPs. It also shows how centrally softening water at a drinking water plant can drastically reduce chloride loading to WWTPs by making home-based softening unnecessary. The benefit of chloride reduction from centralized softening can be realized by combining centralized softening with removal of home-based softeners.

## Reducing chloride: Treatment options, technologies and costs

The ultimate goal for a candidate solution to the chloride problem is to ensure that Minnesota's waters are meeting the relevant WQS to protect aquatic life from the adverse impact of chloride. This requires solving the key problem of sufficient chloride reduction to enable WWTPs to meet their chloride WQBELs at reasonable cost. As home-based softening

**Table 1. Summary of chloride loading sources to WWTPs.**

|  | Alexandria, MN (2014) | | Morris, MN (2014) | | Madison, WI (2016) | |
| --- | --- | --- | --- | --- | --- | --- |
| Chloride Loading Categories | Concentration | Load | Concentration | Load | Concentration | Load |
|  | mg/L | % | mg/L | % | mg/L | % |
| Source Water | 78 | 11% | 4 | 0.50% | 34 | 8% |
| Industrial/Commercial (Ion-exchange) | 121 | 17% | 158 | 19% | 77 | 18% |
| Residential (Non-ion-exchange) | 50 | 7% | 46 | 5.50% | 34 | 8% |
| Residential (Ion-exchange) | 463 | 65% | 672 | 81% | 245 | 57% |
| Road Salt Infiltration | NA | NA | NA | NA | 30 | 7% |
| Hauled Septage | NA | NA | NA | NA | 9 | 2% |
| Average WWTP Effluent | 712 | | 830 | | 430 | |

Data sources for Table 1 –Alexandria and Morris, Minnesota: MPCA Permit Data, Madison, Wisconsin: [17]

contributes significant chloride loading to WWTPs, a candidate solution requires to solve the additional problem of provision of an alternative to home-based softening.

Some may argue that it is simply too expensive to meet certain (lower) levels of chloride discharge in wastewater. In cases where compliance with a standard is likely to cause social and economic hardship, discharging facilities may apply for a variance. A variance is a temporary change to a water quality standard [53]. In Minnesota, chloride variances have included requirements for pollutant minimization plans to assess the most appropriate pollutant reduction strategies and technologies to reduce chloride discharges. At the end of a variance's term, WWTPs may reapply to the state and be evaluated for a subsequent variance. A variance could be a cost-saving option for WWTPs who cannot affordably comply with their chloride limits and may be used in combination with a solution to the chloride problem, such as centralized softening.

An effective solution to the chloride problem must be both environmentally-effective and cost-effective. Determining a solution therefore requires a cost-based comparison of available engineering alternatives based on treatment costs and consequences to society (and the environment). A snapshot of the chloride problem, its impacts, and potential solutions is presented in Fig 5.

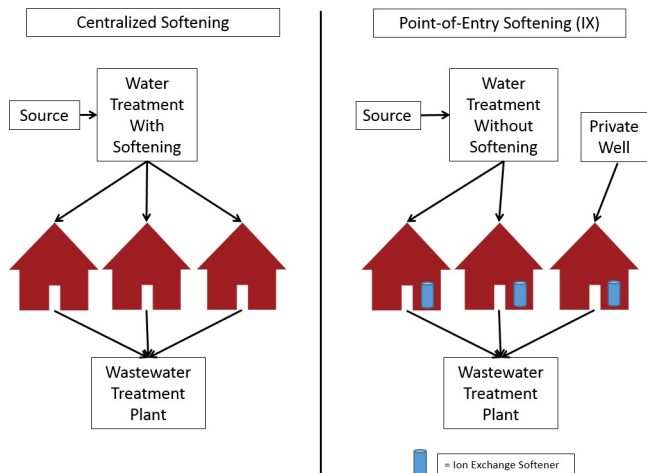

**Fig 4. Chloride loading to WWTPs from home-based softener use.**

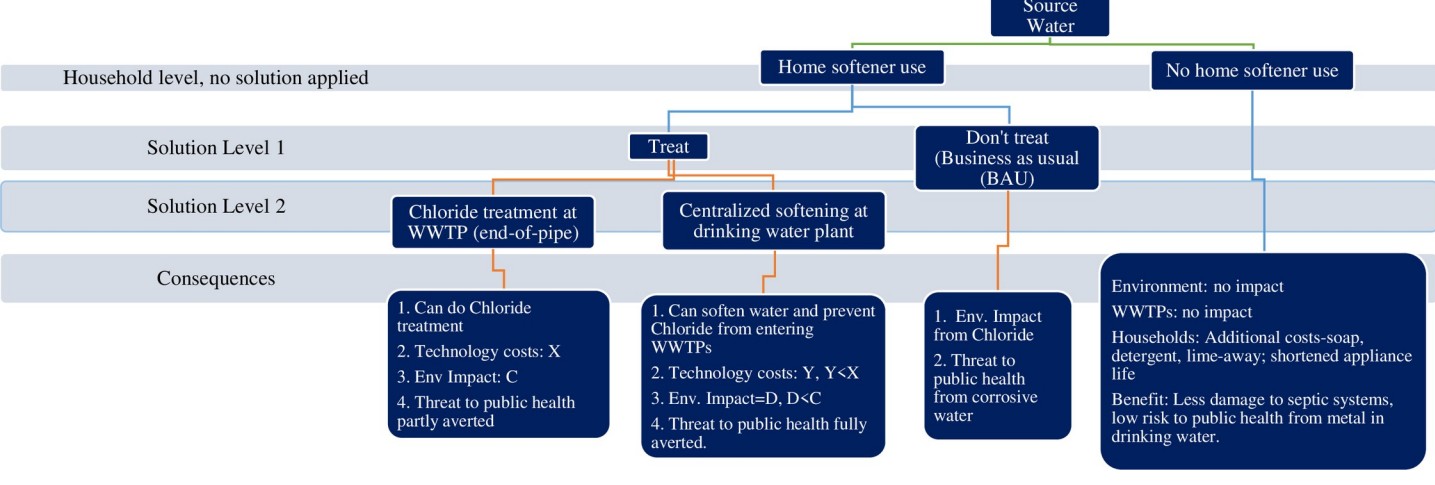

**Fig 5. Snapshot of the chloride problem.**

Detailed information on possible management options and treatment technologies available under each potential solution is shown in Table 2. We consider and compare the costs of two key candidate solutions: (1) shifting to centralized softening at a drinking water plant with removal of all home-based water softeners, and (2) continuing with home-based softening and treating chloride end-of-pipe at WWTPs. These solutions are illustrated under 'Treat' in Fig 5 and in column 2 of Table 2 below. We also include a baseline solution or Business as Usual (BAU) option, which appears under 'Don't treat' in Fig 5.

Although Table 2 lists multiple options for centralized softening, we consider only two in more detail: CS-RO and CS-Lime. These two technologies were found to be most suitable, based on meeting three criteria: 1) allowing WWTPs to meet chloride WQS; (2) feasibility with respect to technology and implementation; and (3) reasonability of cost. We do not consider centralized ion-exchange softening in this analysis because it will not allow WWTPs to meet effluent limits established to ensure attainment of the chloride WQS. For comparison, these suitable CS alternatives and the WWTP chloride treatment alternative (RO effluent—Concentrate crystallized/evaporated), all highlighted in green in column 3 (Alternatives) of Table 2, appear as the two options under 'Treat' in Fig 5.

Among Minnesota communities that have already adopted centralized softening, several have adopted RO or lime-softening. Some have moved to centralized ion-exchange softening, which does result in slightly lower level of chloride loading, compared to home-based softeners, but it is generally not sufficient to be an effective candidate strategy for chloride management and therefore is not considered further. (See Fig 6 for information on which Minnesota communities have adopted centralized softening.)

## Solution strategies and treatment options

We examine three main solution strategies, and options within these solution strategies as presented in Fig 5. As noted before, we have the baseline or BAU strategy, shown under 'Don't treat', following 'Home softener use' in Fig 5. In this scenario, water is softened as needed at the point of use. The baseline strategy does not include any management of chloride load from home-based softening to WWTPs, leading WWTPs with effluent limits facing penalties for chloride violations unless they apply and are approved for a variance.

**Table 2. Chloride and water softening: Options and treatment technologies.**

| | | Alternative | WWTP chloride reductions possible? | Ability to bring WWTP into chloride compliance (~230 mg/L)? | Technical feasibility | Implementation feasibility | Estimated relative cost |
|---|---|---|---|---|---|---|---|
| **Reduce chloride loading to WWTP** | **Drinking water source reduction** | Centralized Lime − softening | Yes | Likely* | Yes | Feasible | High |
| | | Centralized RO − softening | Yes | Likely* | Yes | Feasible | High |
| | | Centralized ion exchange softening | Yes | Unlikely | Yes | Not Feasible | High |
| | | Ferric chloride—> Ferric sulfate | Yes | Unlikely | Yes | Feasible | Low |
| | **Upgrade Water Softeners** | Upgrade to high salt efficiency Point-of-entry softeners | Yes | Unlikely | Yes | Feasible | Medium |
| | | Upgrade industry to high efficiency softeners | Yes | Unlikely | Yes | Feasible | Medium |
| | | Outlaw ion exchange point-of-entry water softeners | Yes | Likely | Yes | Not Feasible | Medium |
| | | Create softener column exchange and Collection Program | Yes | Likely | Yes | Not Feasible | High |
| | | Switch to non-ion exchange softeners | Yes | Likely | No | Feasible yet Unproven | Medium |
| | | Increase residential softening target | Yes | Unlikely | Yes | Not Feasible | Medium |
| **Treat chloride at WWTP** | **WWTP chloride treatment** | RO effluent—Concentrate discharged to surface water | Yes | Likely | No | Not Feasible (Permitting) | High |
| | | RO effluent − Concentrate crystalized/evaporated | Yes | Likely | Yes | Not Feasible (Energy) | Very High |
| | | RO effluent—Concentrate deep well injection | Yes | Likely | No | Illegal | Very High |
| | | Chlorination to UV disinfection | Yes | Unlikely | Yes | Feasible | Medium |
| | | Ferric chloride to ferric sulfate | Yes | Unlikely | Yes | Feasible | Low |
| | | Chloride precipitation with silver nitrate | Yes | Possible | Yes | Not Feasible | Very High |
| | | Chloride anion exchange | Yes | Possible | No | Not Feasible (Untested) | Very High |
| | | Electrodialysis | Yes | Possible | Yes | Feasible | High |
| | | Any biological treatment process | No | Impossible | No | Not Feasible | NA |

*If all residential wells eliminated and in-home softeners disconnected

Compiled by MPCA staff based on engineering research findings

Note: The green highlighting in Table 2 indicates options meeting criteria in columns 5, 6, and 7 (where 7 is implementation feasibility based on permitting), and hence considered in analysis.

We note that while industries also use ion-exchange softeners, their contribution to WWTP chloride loading is on average two to three times lower, compared to home-based ion-exchange softeners. The solutions strategies examined here are intended as a substitute for the use of home-based water softeners by households. In this context, we also considered a few options that focused on managing the landscape of home-based softening, such as: prohibiting the use of home-based softeners, upgrading to **high efficiency home-based softeners** (for example, ion-exchange softeners that work on-demand, and those that offer brine reclamation) in homes and industrially, switching to non-ion-exchange softeners, managing the level to which water is softened at home, and a softener column collection/exchange program, which would enable the recharging of used home-softener columns and reclamation of the brine used in the collection center avoiding its discharge to a WWTP [18].

As Table 2 shows, none of the above management options for home-based softeners are generally workable as standalone solutions, either because they will not sufficiently reduce chloride loading to WWTPs or because they are infeasible to build or implement. For example, prohibiting home-based softeners without an alternative softening option is infeasible as soft water is

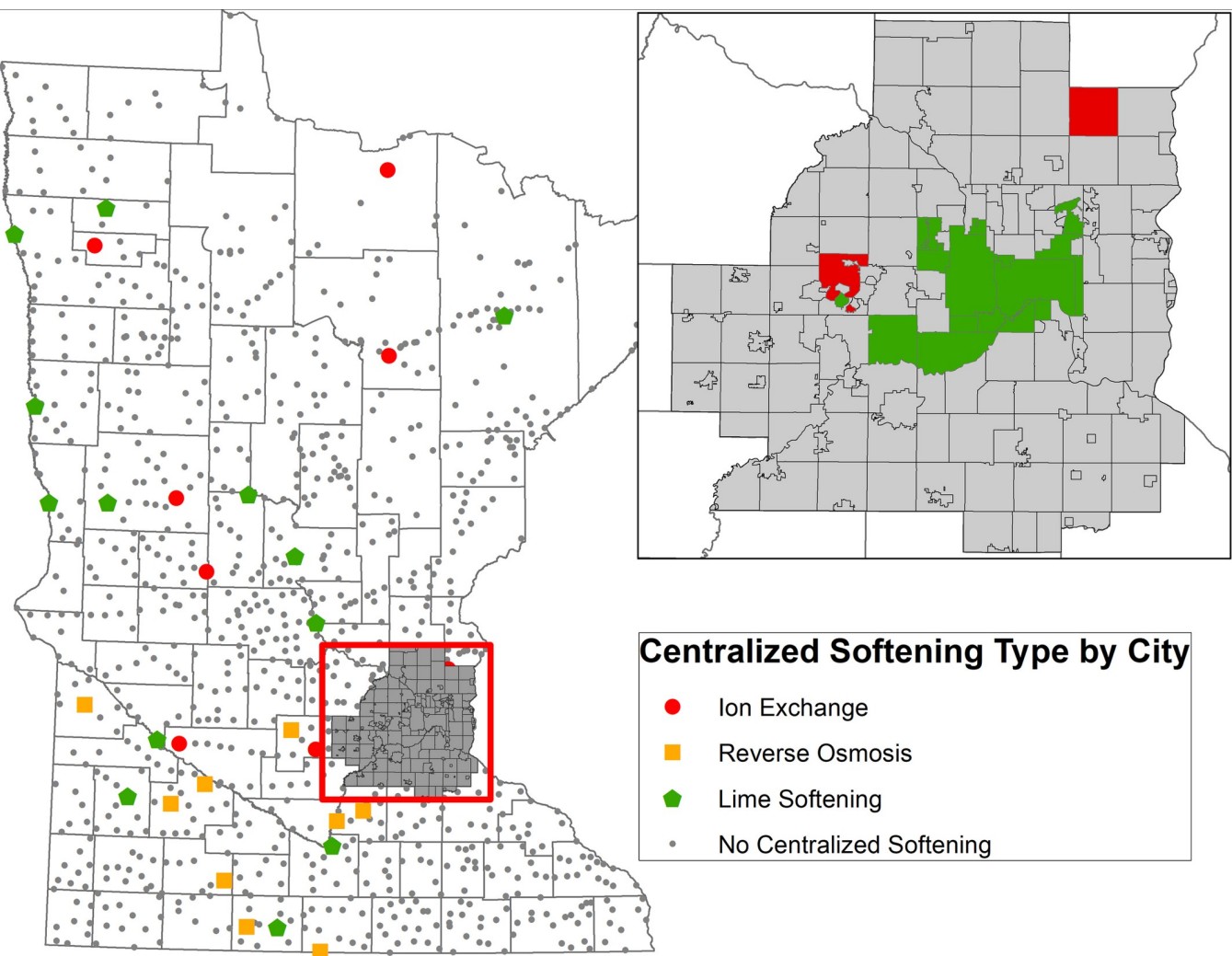

**Fig 6. Distribution of centralized softening technology in Minnesota communities.** Reprinted from the original map image of Fig 6 under a CC BY license, with permission from Steven Weiss, Supervisor-Effluent Limits Unit, Minnesota Pollution Control Agency, original copyright 2020.

necessary for daily activities and smooth running of appliances. Requiring no use of home-based softeners or some specific type of softener may also be politically difficult. Similarly, a softener column exchange program would require weekly trips by water treatment professionals to collect the spent water softener columns from households, treat them with brine to regenerate the resin and replace the treated columns back in households, and reclaiming the spent brine by treatment, entailing not only uncertain and potentially high costs but also infeasible implementation [18]. Consequently, these choices are not analyzed as candidate solutions in this paper.

The two main candidate solutions *either* require centralized water softening at the drinking water plant (see Fig 5, the box marked 'Centralized softening at drinking water plant', following 'Treat'; we only consider drinking water plants that are centralized i.e. water is piped to these facilities for treatment before distribution), combined with removal of home-based softeners, *or* require WWTPs to treat chloride end-of-pipe (see Fig 5, the box marked 'Chloride treatment at WWTP (end-of-pipe)', following 'Treat').

In the case of centralized softening, the initial capital and operation and maintenance (O&M) costs of the softening technology used will be paid by the community's drinking water operations, and a part of the costs will be passed through to users in higher monthly user fees. These costs may include hiring of skilled operators, and chemical usage for corrosion control and disinfection [54]. The level of softening in CS is typically 4–5 gpg of hardness, which is moderately hard water and should not pose a significant threat from corrosion. Even so, some level of chemical usage for corrosion control may be required to ensure the finished water is safe for public distribution. Corrosion control chemicals are typically phosphates, which could pose a concern of potential infection from microbial growth, and consequently need further treatment for disinfection. Based on research, CS technologies have an added benefit of controlling organic compounds, algae, bacteria and viruses, and could reduce the need for chemicals used for disinfection [18].

In order to take advantage of centrally softened water, a majority of homes will need to be connected to city water. This may require installing supply pipes and connections, in which case those installation costs need to be accounted for. Once centralized softening is installed and operational, there will be the cost of removing home-based softeners, for which options (which could range from incentives to regulatory requirements) need to account for political will and public preferences. The Minnesota Pollution Control Agency (MPCA) has a variety of resources to help communities manage softener removal costs and enforcement options, as discussed later in this section. Finally, the by-product emerging from the softening process will have higher hardness compared to source water, which based on the specific centralized softening technology used, could have additional costs and consequences, which are discussed later.

As Table 2 shows, there are four technologies available at the drinking water plant to reduce chloride loading to WWTPs. First is to change the technology for purifying water by replacing the ferric chloride used in the purification process. While this will reduce chloride, it will generally not achieve sufficient chloride reductions. This conclusion holds for the majority of cities. A small percentage of Minnesota cities (~10%) may be able to achieve sufficient chloride reductions by alternative means such as optimizing home-based softeners citywide, but the likelihood of complying with chloride limits by optimizing softeners is low. This change also does not impact softening. Three technologies soften water: CS-RO, CS-Lime, and CS-ion-exchange. We noted earlier that CS-ion-exchange will generally not be a successful chloride reduction strategy. Therefore we are left with CS-RO and CS-Lime as feasible technologies that have the potential to sufficiently reduce chloride loading.

In CS-RO, RO filtration is used to capture and remove the ions contributing to the water's hardness. In CS-Lime, hydrated lime is used to soften water by precipitating out the ions contributing to the water's hardness. Both technologies result in softened water (permeate) that is distributed to households and a by-product (concentrate) that is very hard–as it contains the ions that were removed from the source water. Both options require some additional management actions that impact the implementation cost. In the case of CS-RO, the by-product (concentrate) can be sent to WWTPs as shown by Fig 7 [18], likely without additional cost. Because it is then mixed with the now-centrally softened water that has been used in homes, the resulting effluent has similar hardness as the source water, thus preventing the potential environmental impact of reducing hardness ions [5, 4]. However, the permeate may in some cases require additional treatment with corrosion inhibitors to manage any potential public health risk from lead and copper leaching in the distribution system [55].

In the case of CS-Lime, the by-product is a lime sludge residue, mainly calcium carbonate. This could either be disposed of at a landfill for a cost paid by the drinking water plant, or sold for a variety of potential uses including in wastewater treatment, construction, and agriculture

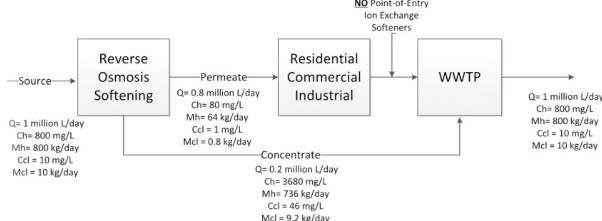

**Fig 7. Stylized diagram of the centralized RO-softening process.** Note on acronyms: Q = Wastewater flow. Ch = Concentration of hardness. Mh = Mass rate of hardness. Ccl = Concentration of chloride. Mcl = Mass rate of chloride.

(fertilizer) [56, 57]. Some of these uses may have associated environmental concerns such as groundwater contamination [58] and/or innovational challenges such as replacing cement with lime sludge to lower the carbon footprint of the concrete production process [59]. However, having a range of choices for a residual product is useful to drinking water treatment facilities [60].

Additional city-specific conditions will impact the choice of the best centralized softening system. CS-RO and CS-Lime are both high cost technologies but they differ in how their capital costs vary with project scale as shown by Fig 8. Based solely on capital costs, CS-RO may be more cost-effective for small to moderate communities serving up to 9000, while CS-Lime might be a better fit for larger communities.

The effectiveness of RO filtration may vary based on water chemistry. This is because natural organic matter and sediments found in surface water require investment in complex engineering design for RO filtration to work which increases cost. For this reason, CS-Lime may be more suited for cities that use surface water rather than groundwater. Apart from removing hardness, both CS-RO and CS-Lime are capable of removing radionuclides such as uranium and radium from drinking water as well as other toxicants such as nitrate, arsenic, iron, and manganese [55, 60]. Both technologies are also capable of reducing other 'salty parameters' such as total dissolved solids (TDS) and specific conductivity, but CS-RO may not be able to achieve sufficient reductions for compliance with any WQBELs for these parameters, when source water has elevated mineral content, as shown in Table 3, adapted from [18]. Moreover, CS-RO may not be as water efficient as CS-Lime and therefore CS-Lime may be a better choice for cities with concerns about sufficient water supply. These additional differences will help inform application based on community WWTP needs, funding, and source water conditions.

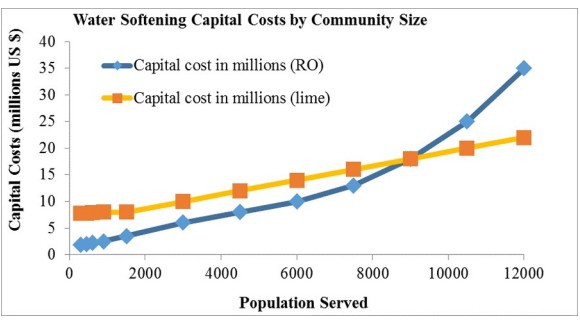

**Fig 8. Capital costs of treatment technologies for centralized softening.** Prepared by Bolton and Menk, Inc. for MPCA use, 2018.

**Table 3. Water softening options compared on RP* for parameters when source water also has RP.**

| | RP at WWTP | | | |
|---|---|---|---|---|
| | *Source* | *ion-exchange* | *Lime** | *RO** |
| **Chloride** | No | Yes | No | No |
| **TDS** | Yes | Yes | No | Yes |
| **Specific Conductivity** | Yes | Yes | No | Yes |
| **Hardness** | Yes | Yes | No | Yes |
| **Alkalinity** | Yes | Yes | No | Yes |

*RP means a facility has **Reasonable Potential** to exceed their effluent limit for the given pollutant in column 1 in Table 3

Implementing either of these technologies has the potential to significantly improve or fully resolve a municipal WWTP's chloride compliance issues. To do so, however, application of either of these technologies must be coupled with removal of home-based softeners from the city/municipal WWTP service area, for which a softener removal cost of $2000 for removing a standard point-of-entry water softener including removal fees, is accounted for in our analysis.

While technology costs of centralized softening are high, it is an efficient solution, which avoids additional chloride loading to WWTPs, compared to the alternative option, which allows it through home-softener use, and then treats it with significantly more expensive end-of-pipe treatment. Furthermore, through centralized softening, additional gains from economies of scale and cost-effectiveness are possible from consolidating softening treatment into the centrally located drinking water plant for a given community, allowing potential cost-sharing between communities, and avoiding purchase, installation, and O&M costs of home-based softeners for users. This can be seen in the experience of several cities with centralized softening, such as Bloomington and Morris. For example, an analysis by the city of Bloomington found that users would save about $30 a month on their drinking water costs through CS-Lime over use of point-of-entry home ion-exchange softeners [18]. Moreover, home ion-exchange softeners are often poorly maintained and users often like the idea of not having to maintain another appliance in their house. Finally, as noted before and documented by current research, centralized softening has a climate change benefit over ion-exchange softening, which implies higher environmental costs of the BAU option and the end-of-pipe treatment options, both of which allow ion-exchange softening [48].

It is worth noting that the direct costs of centralized softening may be further reduced/delayed by applying for public infrastructure funding, applying for MPCA resources directed at removing softeners such as loans and grants, and/or applying for a variance while the centralized softening plant or treatment technology is being built and implemented.

Ten facilities have applied for variances so far. The MPCA has several resources that communities may use to manage home-based softeners through various means, including removal. For example, apart from providing information on efficient salt management [61], the MPCA offers zero interest loans to remove softeners through the Clean Water Partnership (CWP) loan program [62]. The CWP program can also be used to minimize the use of home-based softeners, and either upgrade, optimize or implement their removal from households through financial incentives such as softener rebate programs. The agency recently developed guidelines for softener removal based on a softener rebate grant program [63] and has about $200,000 allocated for this purpose.

Additionally, recent legislative changes [64, 65] that allowed cities access to two infrastructure grant and loan programs, for drinking water infrastructure projects: the Point source implementation Grant (PSIG) and the Water Infrastructure Fund (WIF), for the first time in

2017, and also increased the total amount of potential support from $3 million to $7 million, have substantially increased funding for drinking water projects (see Fig 9 [66]).

PSIG and WIF are jointly administered by the Public Facilities Authority (PFA) [67] and our agency (MPCA), and issue grants and low-interest loans to cities annually for water infrastructure projects, based on a variety of factors related to project funding needs including population, median household income (MHI), age and condition of existing infrastructure, and the existence of effluent limits for specific pollutants such as chloride faced by their wastewater treatment plants. Prior to these amendments, PSIG and WIF were mainly accessible by wastewater projects. The idea behind the amendments was to provide cities with more flexibility in attaining clean water goals by including drinking water infrastructure projects in their application. As centralized softening will have to be installed at drinking water plants to realize the benefits to water quality from reduced chloride pollution as well as provide an alternative way to soften water, these legislations are particularly helpful to cities for corresponding upgrades needed to their drinking water infrastructure. The PSIG program is specifically designed to help communities meet effluent limits on their wastewater systems and contributes up to 80% of the total project costs with a cap of $7 million [68], while the WIF is a long term low interest loan program, which issues loans to cities for infrastructure projects, while protecting ratepayers by capping their water bills to at 1.2%-1.4% of MHI through grants based on affordability [69]. Consequently access to these programs opened up substantial new funds for communities directed towards needed drinking water infrastructure projects, as shown in Fig 9. This increased funding is likely to continue in the future as the legislations supporting them are permanently in the state statute. Because consideration of chloride treatment and management needs is quite recent, very few cities have applied and benefited from these funds for projects related to chloride. However, this is a familiar funding source to many local governments, and many infrastructure projects have been funded through PFA grants.

We will present results on detailed cost comparisons of these technologies accounting for softener costs and related factors at the city and user level in the next section. While centralized softening is not an inexpensive technology, if current conditions with high usage of home-based softeners continues, the only solution for meeting effluent limits is removing chloride in wastewater through additional treatment by WWTPs. This end-of-pipe chloride treatment solution is shown by the box marked 'Chloride treatment at WWTP (end of pipe)' option following 'Treat' in Fig 5. Table 2 lists nine technology choices for treatment of chloride at the

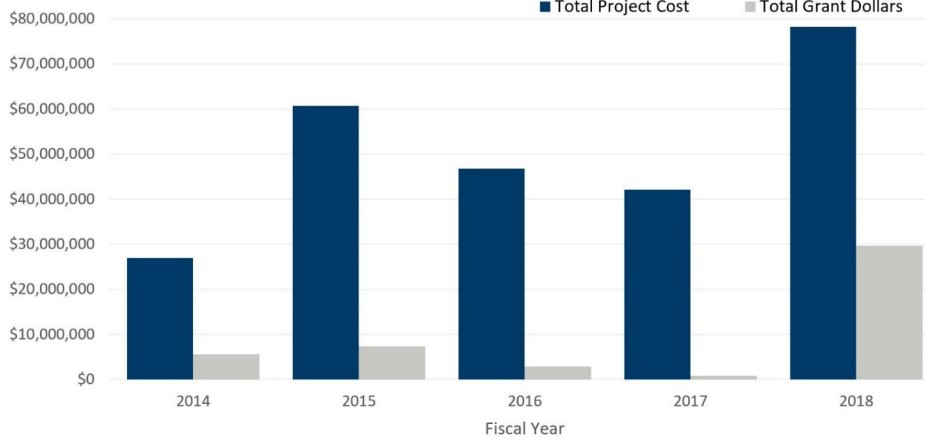

**Fig 9. Grant dollars and total costs for drinking water infrastructure.** Source: Drinking Water Annual Report, 2018, Minnesota Department of Health [60].

WWTP, three of which are likely to result in facilities meeting their specified effluent limits. We only consider RO effluent with EC in this analysis because it is the only technically feasible option among these three choices. If this option is implemented, the capital and O&M costs of the technology will be borne by the WWTP and will be passed on to the public as higher user fees. The costs of end-of-pipe treatment using RO-EC include the cost of the RO filter; the engineering cost of achieving the permeate (desired filtered wastewater) based on the initial water hardness; the cost of evaporating off the concentrated brine reject; the costs of disposing of the resulting salt crystals; and the cost of additional salts that need to be added to the permeate to stabilize the pH of the resulting effluent to levels that are not toxic to aquatic life.

The costs of any construction at a wastewater facility are generally borne by the municipality and passed through, at least partially, to those who use the system. The user fees resulting from installation of end-of-pipe treatment at the WWTP will likely be higher compared to those from centralized softening. The costs of implementing end-of-pipe technologies are also higher than centralized softening, and for certain communities the latter has the additional potential of cost-sharing among a larger population if there is collaboration between cities through development of shared drinking water plants. Based on technology capital costs and using the 2% community MHI threshold provided by EPA, Fig 10 shows that this end-of-pipe chloride treatment option is significantly more expensive compared to centralized softening, unaffordable to most communities, and is particularly cost-prohibitive for small communities in Minnesota.

Moreover, this option is inefficient as it does not combine the water-softening problem with the WWTP chloride compliance problem, so that the public will continue to bear the cost of buying, operating, maintaining, and replacing home-based softeners, apart from paying higher user rates for chloride treatment. It also does not consolidate the softening problem at a central treatment plant so potential additional benefits from economies of scale or consolidation are lost. Furthermore, it does not resolve the impact from **Subsurface Sewage Treatment Systems (SSTS)** [70], which are not able to treat chloride and would therefore still discharge it and potentially impact nearby groundwater or surface waters. Also, it will not sufficiently mitigate the potential public health threat of metal leaching from interior pipes and fixtures due to additional chloride contamination of source water from home-based softening.

## Materials and methods

The main objective of this paper is to examine and compare the three candidate solutions to Minnesota's chloride and water softening problems. The method followed to achieve this objective was to estimate annual costs of each of these solutions as well as costs incurred over

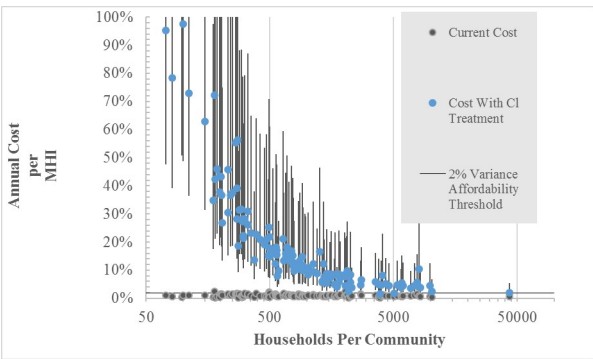

**Fig 10. Capital costs of end-of-pipe chloride treatment.**

the lifetime of a home-based softener in terms of Net Present Value (NPV), at user and city-level for 84 Minnesota cities with matching data on drinking water plants and WWTPs. NPV denotes the current discounted value of a future sum of money or stream of cash flows over time, given a specified rate of return and discount rate [71]. As costs of new technology are incurred immediately after adoption, we present them in terms of annualized values for the current period, based on a 20-year loan. To compare the future costs of these technologies, we also present them in terms of the current discounted worth of future costs over the lifetime of a home-based softener (typically 8–10 years), using NPV. The annualized costs reflect all expenses in the current period and do not include discounting. Both versions are presented for completeness.

Since the centralized softening solution is being considered to avoid additional expensive chloride treatment at WWTPs, we only considered WWTPs that will have to install chloride treatment in the absence of the centralized softening solution, i.e. those WWTPs that have reasonable potential (RP) to violate their chloride limits in the current situation. The drinking water plants included in our analyses are those that can provide centralized softening to the municipalities served by WWTPs with RP for chloride. Drinking water plants may draw water from one or more sources, including surface water such as rivers and lakes, and groundwater sources, such as wells drawing from aquifers. Based on the sources, location, conditions, and available planning and infrastructure, cities may treat water in a water treatment facility before distribution (common for surface water systems), or opt for minimal or no treatment before water is supplied to users (usually only for groundwater-based systems) [72]. For example, Minneapolis draws mainly from the Mississippi River and does extensive treatment [73], while Rochester draws from groundwater and has a distributed network of wells from which water is supplied with minimal treatment [74].

In our analysis, we focused on cities with WWTPs that have RP or limits for chloride and drinking water plants with single, centrally located facilities with treatment and distribution to users, as they are the best potential candidates for applying centralized softening. We started our list of candidate drinking water plants based on 116 regulated drinking water plants with discharge permits out of more than 600 statewide, as drinking water plants with discharge permits may be connected to single treatment centers. This is because the MPCA requires water treatment discharge permits to discharge by products from water treatment, and implementing centralized softening could cause some facilities to dispose to surface water and therefore require a permit.

There are more than 500 municipal WWTPs discharging to surface waters in the state. One hundred and fifty (~29%, i.e., a large enough sample) of these, are monitored for chloride [75], and 96 of the monitored facilities (~64%) have reasonable potential (RP) to exceed chloride limits (Fig 11). Eight (~5%) facilities already have chloride limits, and approximately 100 facilities with RP are likely to get permit limits in the near future. A city with a WWTP having chloride limits or RP and a drinking water plant that is a candidate for centralized softening is considered as a data point in our analysis. Based on this criterion and data availability, our final dataset contained 84 cities. We discuss the details about factors considered in the analysis, data sources, and methods in the rest of this section.

## Factors considered

We estimated the costs of three potential solutions: (1) Business as Usual (BAU), i.e. no chloride reduction, (2) centralized softening with removal of home-based softeners, and (3) treatment of chloride at WWTPs without removal of home-based softeners. The estimation steps required accounting for all of the applicable costs for each solution. The BAU case (and the

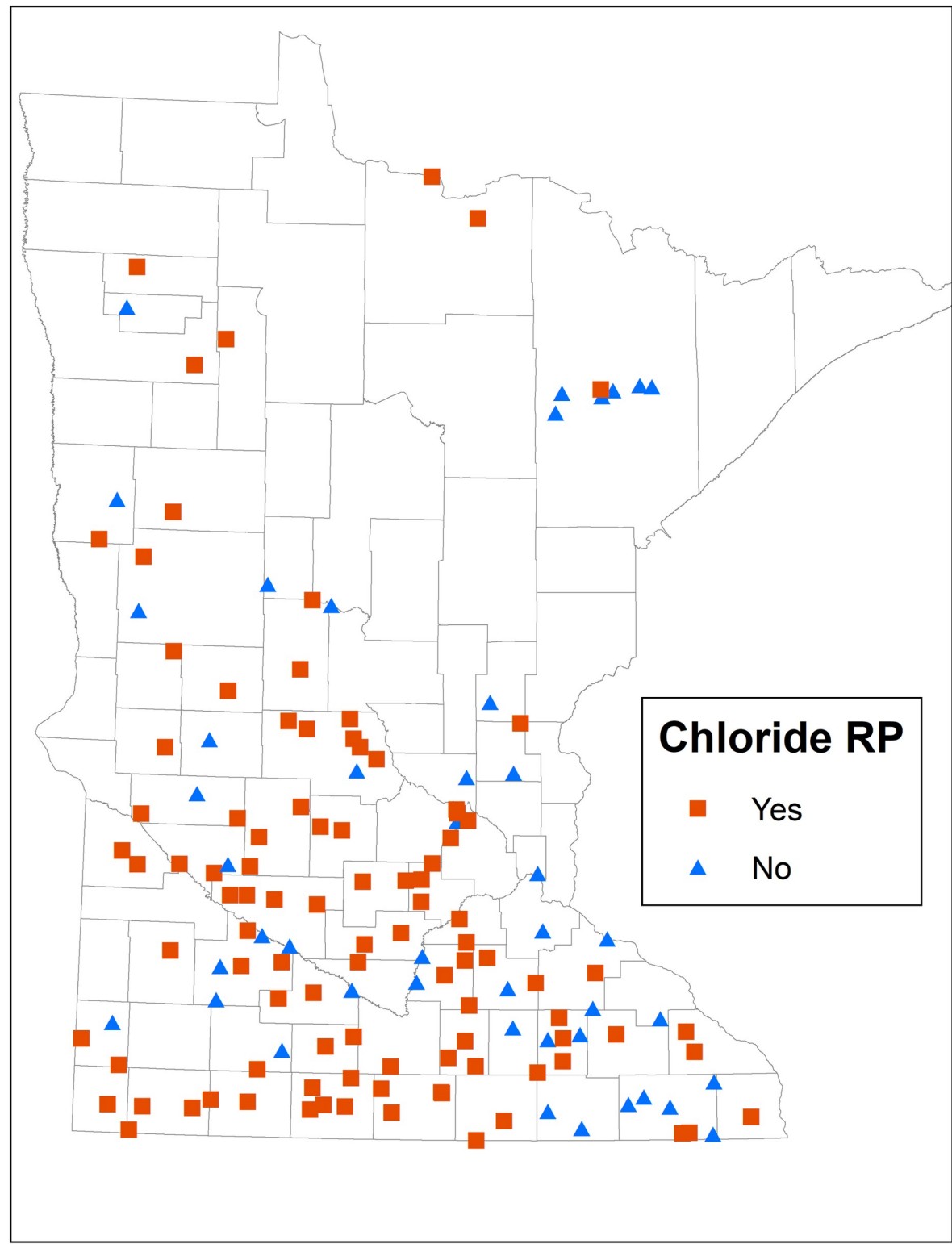

**Fig 11. Prevalence of Minnesota communities with WWTPs having chloride RPs.** Reprinted from the original map image of Fig 11 under a CC BY license, with permission from Steven Weiss, Supervisor-Effluent Limits Unit, Minnesota Pollution Control Agency, original copyright 2020.

end-of-pipe treatment case) accounted for the purchase of a new home-based ion-exchange softener at $2,048, and its maintenance cost at the household end, and the existing O&M costs and liabilities accruing at the drinking water plant and the local WWTP for the corresponding city. The centralized softening (CS) solution for Lime and RO accounted for the capital costs and O&M costs of these technology options and any liabilities at the drinking water plant assuming the construction of a new CS plant and 10-hour working days for operators [18]. As our O&M costs evolve based on plant size and annual flow, parameters which also determine chemical usage, the CS solutions implicitly accounted for any additional costs of chemical usage [53]. It also accounted for the existing O&M costs and liabilities at the WWTP, and the cost of removing softeners from homes. The conservative assumption of building a new plant allows room for any additional costs, for example for hiring specialized workers, if needed. The RO-EC end-of-pipe solution accounted for the capital and O&M costs of treating chloride at the WWTP combined with any liabilities and the existing O&M costs and liabilities at the drinking water plant.

We were not able to account for certain factors due to scope and insufficient data and information. These factors included additional infrastructure costs for potential new household connections to a city water supply, and potential cost reductions that could be had from applying for state and federal infrastructure loans and grants such as WIF and PSIG, for the centralized softening and end-of-pipe treatment solutions, and potential cost reductions for homeowners from rebates for home softener removal owing to chloride being a new pollutant being addressed and consequent current unavailability of data on these parameters. We also did not aim to quantify the environmental costs of chloride pollution as a part of this paper. The implications of these unaccounted factors, and other related assumptions on costs, are discussed in the Results and Discussion section.

## Data

The data for this project was collected from a variety of sources. Most data were at community/city level. Data on municipal WWTPs with RP, were obtained from MPCA internal calculations in 2019. Data related to municipal drinking water plants were obtained from the Minnesota Department of Health, based on geospatial data on Drinking Water Supply Management Areas [76]. Data on capital and O&M costs for the three technologies considered: CS-RO, CS-Lime, and RO-EC, were obtained from the engineering consulting firms Bolton & Menk Inc. and Barr Engineering Company [77]. Data on existing O&M costs and liabilities for drinking water plants and WWTPs were obtained from city financial reports maintained by the Minnesota State Auditor's office [78]. Costs of purchasing and maintaining home-based softeners and removing them from households were determined based on average market costs and/or reasonable estimates of local personnel costs. Data on city population and number of households were obtained from the Minnesota State Demographic Center [79]. Data on city median household income was obtained from the American Community Survey program of the US Census Bureau [80].

## Method

We compared the costs of home-based softening and centralized softening in two ways: (1) comparing the annual costs to households of these potential solutions, and (2) comparing the NPVs of these solutions over the usual operating life of a softener (8 to 10 years). Capital costs for new technologies were annualized based on a 20-year loan with a 3.5% interest rate, accounting for the typical operating life of a new plant and current market interest rates. Annualized costs of new technologies were added to other costs that would accrue annually

and affect user fees including drinking water plant and WWTP O&M costs, liabilities, and softener capital and O&M costs, and softener removal charges as applicable. As noted before, we accounted for a standard softener removal cost of $2000, payable by the user or homeowner decommissioning the removal. We did not incorporate explicit rebates into the softener removal cost as the agency's rebate grant program is new and does not have sufficient data on applicants. However, we annualized the total payment to $140.76 per household using a 20-year payback period at a 3.5% interest rate accounting for availability of softener removal assistance such as low interest loans from the agency's CWP loan program [62]. The resulting figures showed annual costs at community level and were divided by community population to provide annual costs at household level. From our results, the median share of the softener removal cost in annual user cost per household is 5.3% for CS-Lime and 8.4% for CS-RO scenarios. We computed NPVs of different solutions based on average softener life and a 3% discount rate to provide a perspective of costs over time.

## Results and discussion

Our results for costs to Minnesota households accruing annually and over the lifetime of residential softeners, are shown in Figs 12 and 13. Annual and monthly costs per household, representing a given year as well as over the lifetime of a softener, for each of the four options are presented in Table 4.

Annual costs for BAU (Fig 12) appear to be higher than the CS-RO option owing to the conservative assumption of the purchase of a new softener, for which annualizing costs is not meaningful due to the low capital costs. This is not an issue in a lifetime cost perspective, as softener costs even out over a longer time horizon as shown in Fig 13 and detailed in Table 4. We include medians to account for outliers in the dataset. It is possible that most households will simply continue to operate their existing softeners.

Annual and lifetime costs at household and city level for selected small, medium and large cities, are shown in Tables 5 and 6. The definitions of small, medium and large cities, are based on relative population size of Minnesota communities in our dataset: Cities with population less than 5000 are considered small, those with population between 5000 and 9000 are considered medium, and those with population exceeding 9000, are large.

Table 7 presents ratios for lifetime costs of the two centralized softening technology options and the end-of-pipe treatment option, relative to the BAU strategy (home-based softening without any chloride management) for these communities.

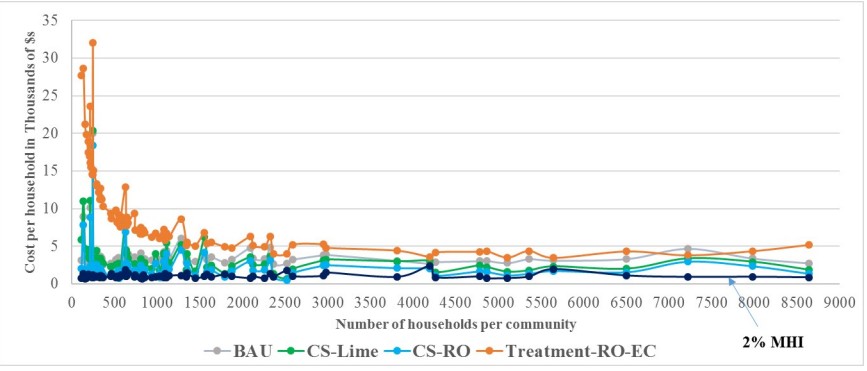

**Fig 12. Annual costs of alternative chloride solutions in Minnesota communities.**

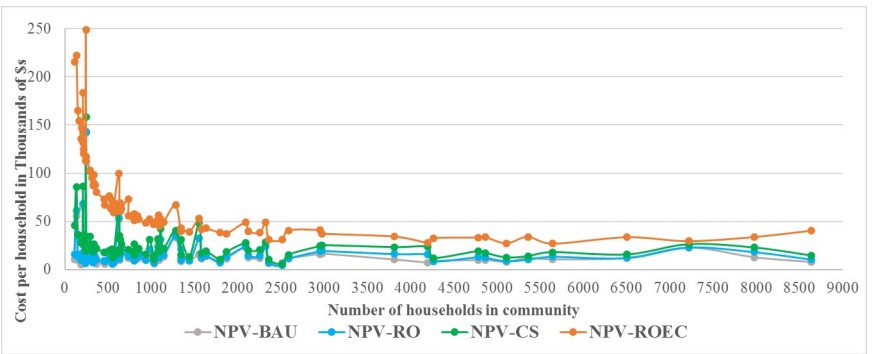

**Fig 13. NPV comparison of alternative chloride solutions over life of home-based softener (8–10 years).**

## Discussion

The results demonstrate that centralized softening is significantly less expensive compared to treating chloride end-of-pipe with an annual average cost ratio of 1:3 for Lime-softening, and 1:4 for RO-softening. For small cities with population up to 5000, this ratio improves for RO-softening at 1:5, and stays unchanged for Lime-softening. We also find that the lifetime costs of centralized RO-softening is on average only 1.1 times that of the BAU strategy, i.e. using home-based softeners and not treating chloride. We find that end-of-pipe treatment for chloride, leaving home-based softeners in place, is the most expensive option the majority of the time. While chloride management technology can be expensive for small communities, as shown by the comparison with the 2% MHI affordability threshold, given available technology options, our results show that centralized softening at a drinking water plant, if implemented, would be the most cost-effective solution for communities that need to address chloride pollution. Considering additional benefits to the environment, public health, and cost savings associated with removal of home-based softeners, which have not been assessed, centralized softening at a drinking water plant would be the best overall solution to manage chloride.

For small to medium sized communities with populations not exceeding 9000, centralized RO-softening would be lower cost compared to centralized lime-softening. This conclusion is applicable to most communities with chloride RP in our dataset, as most are small communities. Some communities in our dataset are either very large and/or have other features related to the corresponding drinking water plant or community characteristics that make this kind of infrastructure shift difficult. For example, in Rochester centralized RO-softening would be more expensive compared to centralized lime-softening; while costs would still make shifting to centralized softening favorable from BAU, there may be infrastructural limitations to such a shift due to the way Rochester's drinking water distribution is designed. Tables 5 and 6 show these differences in annual and lifetime costs at household and city level for selected small,

**Table 4. Annual and monthly user costs in US dollars across all communities.**

| Household level | BAU | | Centralized softening | | | | Home softening | |
|---|---|---|---|---|---|---|---|---|
| | | | CS-RO | | CS-Lime | | RO-EC | |
| | Annual | Monthly | Annual | Monthly | Annual | Monthly | Annual | Monthly |
| Average cost all cities | 3744.70 | 312.06 | 2357.44 | 196.45 | 3391.55 | 282.63 | 9157.92 | 763.16 |
| Annualized average lifetime costs | 1693.31 | 141.11 | 2036.55 | 169.71 | 2929.90 | 244.16 | 7911.37 | 659.28 |
| Median cost all cities | 3260.68 | 271.72 | 1737.21 | 144.77 | 2746.52 | 228.88 | 7158.67 | 596.56 |
| Annualized median lifetime costs | 1275.17 | 106.26 | 1500.75 | 125.06 | 2372.67 | 197.72 | 6184.25 | 515.35 |

**Table 5. Costs of alternatives in ('000) of $s for selected Minnesota cities-household level.**

| Household level | BAU | | Centralized softening | | | | Home softening | |
| | | | CS-RO | | CS-Lime | | RO-EC | |
| | Annual | NPV | Annual | NPV | Annual | NPV | Annual | NPV |
|---|---|---|---|---|---|---|---|---|
| Altura | 3.21 | 11.09 | 1.90 | 14.74 | 4.41 | 34.31 | 18.91 | 147.00 |
| Sherburn | 3.04 | 9.74 | 1.46 | 11.36 | 2.51 | 19.49 | 8.39 | 65.24 |
| Avon | 3.98 | 17.08 | 2.43 | 18.89 | 3.42 | 26.56 | 8.12 | 63.11 |
| Barnesville | 3.04 | 9.77 | 1.59 | 12.38 | 2.37 | 18.43 | 5.82 | 45.25 |
| Pipestone | 4.77 | 23.23 | 3.05 | 23.70 | 3.53 | 27.43 | 6.27 | 48.77 |
| Thief River Falls | 2.90 | 8.70 | 1.09 | 8.51 | 1.53 | 11.93 | 4.19 | 32.59 |
| Worthington | 3.02 | 9.62 | 1.68 | 13.05 | 2.46 | 19.11 | 4.26 | 33.10 |
| Willmar | 3.39 | 12.48 | 2.35 | 18.30 | 2.93 | 22.76 | 4.34 | 33.72 |
| Rochester | 3.27 | 11.58 | 10.87 | 84.49 | 2.52 | 19.61 | 3.59 | 27.93 |

medium, and large cities, based on our data. As noted before, Table 7 presents ratios for lifetime costs of the two centralized softening options and the end-of-pipe treatment option, relative to the BAU strategy, showing that, while centralized softening is expensive, it is significantly less expensive compared to the alternative end-of-pipe treatment solution.

Annual and lifetime costs at the household level for communities with population below 9000 people are shown in Figs 12 and 13. A summary of results across all communities is shown in Table 4 and for selected communities in Tables 5–7. As 74 out of 84 communities in our dataset had populations smaller than 9000, Figs 12 and 13 adequately represent costs across different solutions examined.

We draw several key inferences from the results: First, centralized softening using either Lime or RO technologies is overall a reasonable cost solution compared to end-of-pipe treatment. For most communities, RO-softening is the lower cost option among the centralized softening options, and is on average only 1.1 times as costly as the BAU option over softener lifetime. For larger communities centralized Lime-softening could be more cost-effective as shown by Fig 8. Secondly, end-of-pipe treatment of chloride is the highest cost option in terms of both annual and lifetime comparison with the BAU option. It is on average five times as costly as BAU, three times as costly as Lime-softening, and four times as costly as RO-softening. Thirdly, these ratios become more unfavorable for end-of-pipe treatment in small communities, with end-of-pipe treatment costs on average being five times as high as RO-

**Table 6. Costs of alternatives in ('000) of $s for selected Minnesota cities-city level.**

| City level | BAU | | Centralized softening | | | | Home softening | |
| | | | CS-RO | | CS-Lime | | RO-EC | |
| | Annual | NPV | Annual | NPV | Annual | NPV | Annual | NPV |
|---|---|---|---|---|---|---|---|---|
| Altura | 626.25 | 2,163.45 | 369.70 | 2,874.39 | 860.54 | 6,690.67 | 3,686.80 | 28,664.71 |
| Sherburn | 1,664.23 | 5,335.77 | 800.57 | 6,224.36 | 1,373.40 | 10,678.08 | 4,597.97 | 35,748.98 |
| Avon | 2,607.82 | 11,187.50 | 1,591.55 | 12,374.21 | 2,237.70 | 17,397.94 | 5,316.48 | 41,335.40 |
| Barnesville | 3,326.59 | 10,684.75 | 1,742.44 | 13,547.34 | 2,593.52 | 20,164.53 | 6,366.82 | 49,501.63 |
| Pipestone | 9,993.82 | 48,646.97 | 6,382.97 | 49,627.26 | 7,386.67 | 57,430.88 | 13,135.52 | 102,127.90 |
| Thief River Falls | 12,381.55 | 37,088.70 | 4,666.81 | 36,284.02 | 6,542.42 | 50,867.12 | 17,879.95 | 139,015.59 |
| Worthington | 14,456.57 | 46,034.41 | 8,029.08 | 62,425.56 | 11,753.79 | 91,385.29 | 20,363.34 | 158,324.00 |
| Willmar | 27,047.65 | 99,571.01 | 18,778.70 | 146,003.20 | 23,364.40 | 181,656.80 | 34,611.26 | 269,100.36 |
| Rochester | 156,015.26 | 551,780.71 | 517,858.21 | 4,026,320.57 | 120,189.39 | 934,466.49 | 171,168.91 | 1,330,828.59 |

Table 7. Lifetime costs for selected communities relative to BAU.

| Community | CS-RO to BAU | CS-Lime to BAU | RO-EC to BAU |
|---|---|---|---|
| **Altura** | 1.33 | 3.09 | 13.25 |
| **Sherburn** | 1.17 | 2.00 | 6.70 |
| **Avon** | 1.11 | 1.56 | 3.69 |
| **Barnesville** | 1.27 | 1.89 | 4.63 |
| **Pipestone** | 1.02 | 1.18 | 2.10 |
| **Thief River Falls** | 0.98 | 1.37 | 3.75 |
| **Worthington** | 1.36 | 1.99 | 3.44 |
| **Willmar** | 1.47 | 1.82 | 2.70 |
| **Rochester** | 7.30 | 1.69 | 2.41 |

softening. Table 7 shows how cost-prohibitive end-of-pipe chloride treatment can be for certain communities. While centralized softening technologies could also be expensive for some communities, there are options for potential cost reductions through infrastructure loans and grants and an option to delay costs owing to socioeconomic reasons through a variance.

## Discussion on cost estimates

Our cost estimates for the CS and RO-EC scenarios, should be considered as upper bounds due to several reasons including conservative assumptions on costs, and data limitations. For example, owing to conservative assumptions such as having to account for building new plants to install CS or RO-EC technologies, and not accounting for financial assistance to cities for infrastructure costs, or to users for softener removal costs, due to lack of data, our cost estimates for the CS and RO-EC options overestimate the potential actual costs of these scenarios.

Similarly, based on the scope of this paper, we did not account for the environmental costs of the BAU or the end-of-pipe treatment options. It is important to note, that the BAU scenario particularly, leads to underestimating the total costs to society from continued chloride pollution including potential losses to valuable ecosystem services, i.e. benefits received from nature [81] such as water quality, nutrient cycling, wildlife habitat, recreation, and clean drinking water (which includes the public health costs of excessive chloride in drinking water). There has been little research estimating the environmental costs of chloride pollution. We discuss a few examples from the literature here to provide a perspective on these unaccounted environmental costs of the BAU scenario. Estimates of ecosystem service valuation based on peoples' willingness to pay (WTP) for restoration and indirect estimates through species like freshwater mussels that are sensitive to chloride pollution [29] and also provide ecosystem services [82], provide useful reference on these costs. For example, a study in Brazil found local residents willing to pay increases to their current water bills by 0.81% to 1.25% of their average income, to restore water quality in the Paraíba do Sul River, which was contaminated with industrial and agricultural pollutants over many years [83]. With regard to freshwater mussels, a study in the Kiamichi River in Southeastern Oklahoma, found three ecosystem services: biofiltration, nutrient cycling, and nutrient storage, declined by 60% over a 20-year period in proportion to the decline in freshwater mussels in the river [84]. A survey of stakeholders for the same river found habitat provision and water quality had the highest economic value with annual WTPs per household at $20.34 and $9.59 [85]. Chloride being toxic to fish [3], valuation estimates for fishing are also indicative of potential losses from chloride pollution. The freshwater fishing industry generates $2.4 billion in expenditures annually in Minnesota [86], and $30 billion [87] nationally, so that excessive chloride contamination of surface water could

affect fishing, recreation, and tourism. Chloride also degrades groundwater sources used for drinking water, and the replacement costs of chloride-contaminated drinking water through desalination were found to be substantial, estimated as $3600 per acre foot of drinking water [88]. Groundwater salinity can also result in agricultural losses. For example increasing salinity of irrigation water was associated with declining farmland values in California's Central Valley [89]. The economic costs of the public health crisis in Flint, Michigan, caused by corrosion in the drinking water distribution system from elevated levels of chloride in source water, have been estimated to be more than $360 million in averting expenditures and government spending [90]. The BAU scenario would also have an added environmental loss from a higher carbon footprint compared to centralized softening, which was found to mitigate climate change by 0.11 Million tonnes $CO_2$ eq./year in a life cycle study of multiple softening options in the Netherlands [48].

Therefore, accounting for the environmental and public health costs of the BAU or the end-of-pipe treatment options would increase the total costs to society from these options, and reinforce the benefits of the centralized softening options. Asking and accounting for the public health and environmental impacts of not solving the chloride pollution problem is a logical next step for future research in this topic.

## Conclusions

In this paper we explored the chloride pollution problem in Minnesota, its association with home-based water softening, and compared potential solutions to the problem in terms environmental-effectiveness and cost-effectiveness. We find that centrally softening water at drinking water plants is an environmentally-effective solution that allows WWTPs to comply with chloride WQS and is also reasonable in terms of cost compared to treating chloride end-of-pipe.

While facilities have other options–such as applying for a variance that may last until the cost of chloride removal at the WWTP is feasible–we find that centralized softening is, in many circumstances, likely to be a key component of reducing chloride pollution. Although that discussion is presented here in the context of meeting an effluent limit, it may also be something to consider even if a municipality's WWTP is not yet subject to a limit. Apart from being more cost-effective, the centralized softening options have four more benefits compared to end-of-pipe chloride treatment:

1. They would protect waterbodies from further ionic pollution and corresponding environmental harm to aquatic life by making home-based ion-exchange softeners unnecessary. This feature would provide more protection to waterbodies in the event end-of-pipe treatment is delayed owing to variances and/or if end-of-pipe treatment cannot be utilized by some households (for example, households that are not connected to municipal WWTPs, and have private systems such as SSTS, from which wastes may flow directly to waterbodies).

2. They would prevent any harm to public health from potential leaching of metals from taps due to chloride contamination of source water or groundwater from improperly treated wastewater.

3. They provide an efficient solution by combining the chloride problem and the water softening problem, thereby saving the public money, time, and effort now spent on buying, maintaining, and replacing home-based softeners, as well as on end-of-pipe chloride treatment costs.

4. They could provide additional benefits from the economies of scale of consolidating softening into a centralized drinking water plant for a given community, and have further potential of cost reduction through multiple communities sharing drinking water plants.

Based on our results and these additional benefits, centralized softening is overall the most cost-effective and environmentally sound solution to Minnesota's chloride pollution problem. As discussed before, our findings are applicable nationally as chloride pollution and its environmental and economic impacts are currently affecting many states in the nation. We hope that this paper provides useful information about the potential of centralized softening, to help multiple stakeholders across academia, government, industry, and homeowners, interested in clean water, and the economics and engineering aspects of chloride pollution. We also hope that this paper encourages future research into the specific water quality impacts of chloride pollution and the ecosystem and health benefits possible from preventing it.

## Supporting information

**S1 Data. Data for paper.**
(XLSX)

## Acknowledgments

We are grateful to Brooke C. Asleson, William M. Cole, Meghan Hemken, Catherine Neuschler, and Casey Scott for their insightful comments and suggestions, and help with data collection and formatting, for this paper. We thank Jennifer Holstad for catching typos and helping with formatting the paper. We thank the Minnesota Department of Health for their thoughtful review, comments and suggestions, which improved this paper.

## Author Contributions

**Conceptualization:** Baishali Bakshi, Scott J. Kyser.

**Data curation:** Baishali Bakshi, Elise M. Doucette, Scott J. Kyser.

**Formal analysis:** Baishali Bakshi.

**Investigation:** Baishali Bakshi, Elise M. Doucette, Scott J. Kyser.

**Methodology:** Baishali Bakshi.

**Supervision:** Baishali Bakshi.

**Validation:** Baishali Bakshi, Scott J. Kyser.

**Visualization:** Baishali Bakshi, Scott J. Kyser.

**Writing – original draft:** Baishali Bakshi.

**Writing – review & editing:** Baishali Bakshi, Elise M. Doucette, Scott J. Kyser.

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
