## [Decision Letter · Decision Letter 0]

28 Sep 2020

PONE-D-20-25024

Centralized softening as a solution to chloride pollution: An empirical analysis based on Minnesota cities

PLOS ONE

Dear Dr. Bakshi,

Thank you for submitting your manuscript to PLOS ONE. After careful consideration, we feel that it has merit but does not fully meet PLOS ONE’s publication criteria as it currently stands. Therefore, we invite you to submit a revised version of the manuscript that addresses the points raised during the review process.

We look forward to receiving your revised manuscript.

Kind regards,

Huan Li

Academic Editor

PLOS ONE

Journal Requirements:

2.We note that [Figure(s) 1, 2, 3, 6 and 11] in your submission contain [map/satellite] images which may be copyrighted. All PLOS content is published under the Creative Commons Attribution License (CC BY 4.0), which means that the manuscript, images, and Supporting Information files will be freely available online, and any third party is permitted to access, download, copy, distribute, and use these materials in any way, even commercially, with proper attribution. For these reasons, we cannot publish previously copyrighted maps or satellite images created using proprietary data, such as Google software (Google Maps, Street View, and Earth). For more information, see our copyright guidelines: http://journals.plos.org/plosone/s/licenses-and-copyright.

1.    You may seek permission from the original copyright holder of Figure(s) [1, 2, 3, 6 and 11] to publish the content specifically under the CC BY 4.0 license. 

Reviewers' comments:

Reviewer's Responses to Questions

**Comments to the Author**

1. Is the manuscript technically sound, and do the data support the conclusions?

Reviewer #1: Yes

Reviewer #2: Yes

Reviewer #3: Yes

2. Has the statistical analysis been performed appropriately and rigorously? 

Reviewer #1: Yes

Reviewer #2: Yes

Reviewer #3: Yes

3. Have the authors made all data underlying the findings in their manuscript fully available?

Reviewer #1: Yes

Reviewer #2: Yes

Reviewer #3: Yes

4. Is the manuscript presented in an intelligible fashion and written in standard English?

Reviewer #1: Yes

Reviewer #2: Yes

Reviewer #3: Yes

5. Review Comments to the Author

Reviewer #1: The paper conducts a net present value analysis of four (three treatments, plus a baseline) scenarios regarding ways to deal with the high levels of chlorine loading in Minnesota waters. The paper makes recommendations regarding the most cost-effective scenario using a series of factors including population size (of municipal WWTP), O&M and capital costs of technologies, etc. The results indicate that central softening is a more cost effective solution than treating chloride at the WWTP (end of pipe).

Overall, the paper is well written, the methodology is presented in a clear way and the findings make a compelling argument in favor of a centralized cost-effective solution that contributes to the efficient allocation of resources.

A have a few comments that I believe would help strengthen the methodology and results of the paper.

Major comments:

1) Page 21: The authors state that they did not quantify the environmental costs of chloride pollution. This is a very reasonable choice given the scope and research question of the paper. However, this probably results in an underestimation of the cost of the BAU scenario (assuming that the penalty imposed by the regulator for exceeding chloride levels is a low bound of the true environmental cost of non-compliance). All three treatment scenarios (CS-RO, CS-Lime, RO-EC) appear to have greater lifetime costs compared to BAU (based on the results presented in Table 7 and Figure 13). It is important for the authors to clarify that the BAU estimates represent a lower bound. Again, I acknowledge that it is beyond the scope of the paper to incorporate the environmental cost of high chloride concentrations. That said, it would be useful to consider whether some estimates of the environmental cost of chloride concentrations have already been documented in the literature. If such estimates exist, I’d like to see them discussed (briefly) in the manuscript so that the reader can get a sense of the magnitude of the environmental cost associated with the BAU scenario. At minimum, the analysis needs to stress this limitation in order not to make the BAU solution appear as the most cost-effective one.

2) It would be interesting to clarify what share of the CS-RO and CS-Lime scenarios costs would go towards covering home-softener removal (rebate plus labor cost associated with the removal) vs. capital and O&M costs of technology installation at the water treatment plant. Some of that information is provided in the supporting information .csv file that the authors have made available to the reviewers, but it would be useful to include a summary of that information in the main manuscript.

3) Related to point #2 above, the authors mention (on page 17) that funds are available from a rebate grant program towards retiring existing home-softeners. It would be useful to consider several alternatives for the retirement of existing home-softeners. Different alternatives could have different cost-effectiveness implications. For example, would all households in a given city need to retire their softeners? Or would it be more cost-effective to focus on households with high water consumption? If rebates are provided as an incentive to retire a home-softener, how would each household rebate be determined? I assume rebates will be determined based on the NPV of the remaining lifetime of the home-softener, but perhaps there are other ways. An alternative approach (instead of offering a rebate) would be to offer a discount to a household’s water utility bill over the course of a few months/years. That might help spread out the cost of the retirement investment (the personnel cost of removing the water softener would still need to be paid in the present).

Minor comment:

1) Page 17: Can you provide some more detail as to why grants increased so much in 2018 (Figure 9)? Is that increased spending likely to continue into the future?

Reviewer #2: The paper is generally well-written (although it could use a proofreading, especially for comma use), it is of limited value to anyone outside of Minnesota as written. Without a more extensive attempt to relate this research to more general water quality problems and solutions that would be applicable in other states or countries, this paper reads very much as a local analysis for local water quality benefit.

Reviewer #3: The manuscript “Centralized softening as a solution to chloride pollution: An empirical analysis based on Minnesota cities” by Bakshi et al. is a well written, comprehensive discussion of the potential for centralized water softening as a solution for a challenging problem, chloride pollution. I think the article is in good shape and a useful contribution to the literature (I am less familiar with the engineering side, and it may be well discussed there, but this topic would benefit the discussion on the env. sci. side. I have some suggestions for improvement, listed below.

Figures in general: The figures should be more focused. For example, the national USGS map (Figure 2) could be cited and a map focused on MN/the 5-6 state region replace that map. Use of those maps doesn’t necessarily add much to the paper.

ll 77: Probably better to call “bugs” “insects” or “macroinvertebrates” or the like

ll 99: It would be useful to also report this as a proportion of the population (i.e., not just a proportion of the counts)

ll 113: this is an incomplete sentence

ll 125-126: While I am not necessarily advocating you do this in this manuscript, it would be helpful to include carbon life cycle implications of these scenarios as you continue this sort of analysis.

ll 153: make it clear this is 75% of the residents (just to be scrupulous)

Table 1; Data sources should be cited in the caption so this can stand alone.

Table 2: It’s not clear what the green highlighting means. Can you clarify in the caption?

Table 3: Consider spelling out RP in the caption to help the reader.

Page 21, “Data” section: When these data are pulled from publicly available databases, citations really should be provided.

References: I strongly encourage the use of more original/authoritative sources than Wikipedia. I agree, it’s a tremendous resource, but, for example the grains per gallon is stripped directly from other fundamental information sources such as the “Standard Methods for Analysis of Water and Wastewater.” Citations help those important community efforts. Don’t starve ‘em

Related question:

Do changes in geochem (e.g., a more sulfate dominated mix, in places like the Red River Valley) impact these cost estimates? I.e., does the sulfate foul things?

6. PLOS authors have the option to publish the peer review history of their article (what does this mean?). If published, this will include your full peer review and any attached files.

Reviewer #1: **Yes: **Nikolaos Zirogiannis

Reviewer #2: No

Reviewer #3: No

---

## [Author Response · Author response to Decision Letter 0]

16 Dec 2020

Response to Reviewers for manuscript titled “Centralized softening as a solution to chloride pollution: An empirical analysis based on Minnesota cities

December 14, 2020

Dear editor and reviewers, 

Thank you for reviewing our manuscript and for your helpful comments. This rebuttal letter contains our responses to each of your comments, apart from 1, 2, 3, and 4 from the reviewers. We are happy to note from the comments 1, 2, 3, and 4, that the reviewers find our manuscript to be technically sound, the statistical analysis appropriate and rigorous, the data to be supportive of our conclusions, and made fully available, and the writing to be in standard English and intelligible. Please let us know if you have any question/s.

Sincerely,

Baishali Bakshi (on behalf of the author team)

Response: We have reviewed the Plos-One style and naming requirements and have made applicable changes to our manuscript. These can be seen in the marked-up version called ‘Revised manuscript with track changes’, that we have uploaded. 

2.We note that [Figure(s) 1, 2, 3, 6 and 11] in your submission contain [map/satellite] images which may be copyrighted. All PLOS content is published under the Creative Commons Attribution License (CC BY 4.0), which means that the manuscript, images, and Supporting Information files will be freely available online, and any third party is permitted to access, download, copy, distribute, and use these materials in any way, even commercially, with proper attribution. For these reasons, we cannot publish previously copyrighted maps or satellite images created using proprietary data, such as Google software (Google Maps, Street View, and Earth). For more information, see our copyright guidelines: http://journals.plos.org/plosone/s/licenses-and-copyright.

1. You may seek permission from the original copyright holder of Figure(s) [1, 2, 3, 6 and 11] to publish the content specifically under the CC BY 4.0 license. 

Response for Figure 2: Figure 2, titled ‘Chloride concentrations in urban streams in the US’ is from a USGS report which is in the public domain and does not require copyright permissions. A statement as to this (italicized) is provided below from a USGS representative, Sabra A. Lopez, who has indicated that all material including images from the specific paper falls under the public domain. We have also uploaded the screenshot of the email received from Sabra A. Lopez, which contains this statement. We hope this is sufficient for publishing Figure 2 for Plos-One as part of our manuscript.

Greetings,

Thank you for contacting the USGS. In reference to:

https://pubs.er.usgs.gov/publication/cir1211

Copyright Permission Agreement for Social Media Submissions

By submitting to the U.S. Geological Survey (USGS), the text, image(s), audios, videos or audio-videos (the Work), tagging its accounts, and using the specified hashtag, the Submitter avers that the work is either in the public domain as a work created by a Federal Government employee in their official capacity or that the Submitter is the sole or joint Copyright holder with authority to grant its use. For Copyrighted work, the Submitter grants to the USGS a non-exclusive, fully paid and royalty-free, worldwide rights to publish, reproduce, make derivatives use and to distribute the Work in any media for the duration of the Copyright. If the USGS uses any of the Work in its products or services, it agrees to provide proper notice and attribution to the copyright holder. Submitter hereby warrant and represent that he or she has the full right, power, and authority to grant the permission requested herein and that use of the work will not violate any rights of any third party. Submitter further agrees that if the above representations concerning copyright ownership are determined to be incorrect or false, resulting in the USGS or the U.S. Government being sued for copyright infringement, that Submitter shall indemnify the USGS and/or the U.S. Government for any resulting out-of-pocket expenses arising from defending and/or settling such litigation.

Sabra Lopez

Science Information Services

Office of Communications and Publishing

U.S. Geological Survey

Box 25046, MS 150, DFC

Denver, CO 80225

Voice: 303-202-4581

(888) 275-8747

Response for Figures 1, 3, 6 and 11: Figures 1, 3, 6, and 11 are produced by our agency (all authors are employed by this state government agency), Minnesota Pollution Control Agency. Copyright permissions for use of agency figures is provided by Mr. Steven Weiss, a supervisor with authority to provide copyright permissions for these and any other figures made by our agency. Additionally we have noted that the previous figures were not high-resolution, and have provided higher-resolution replacements to the earlier figures. We have uploaded the copyright form, filled and signed for this purpose. We hope that this is sufficient for publishing these images as part of our manuscript. 

Response: We have reviewed the supporting information guidelines and have included the following caption for our supporting information file: Data for paper, at the end of the manuscript. We do not have any in-text citations for this file in our manuscript. 

5. Review Comments to the Author

Reviewer #1: The paper conducts a net present value analysis of four (three treatments, plus a baseline) scenarios regarding ways to deal with the high levels of chlorine loading in Minnesota waters. The paper makes recommendations regarding the most cost-effective scenario using a series of factors including population size (of municipal WWTP), O&M and capital costs of technologies, etc. The results indicate that central softening is a more cost effective solution than treating chloride at the WWTP (end of pipe).

Overall, the paper is well written, the methodology is presented in a clear way and the findings make a compelling argument in favor of a centralized cost-effective solution that contributes to the efficient allocation of resources.

A have a few comments that I believe would help strengthen the methodology and results of the paper.

Major comments:

1) Page 21: The authors state that they did not quantify the environmental costs of chloride pollution. This is a very reasonable choice given the scope and research question of the paper. However, this probably results in an underestimation of the cost of the BAU scenario (assuming that the penalty imposed by the regulator for exceeding chloride levels is a low bound of the true environmental cost of non-compliance). All three treatment scenarios (CS-RO, CS-Lime, RO-EC) appear to have greater lifetime costs compared to BAU (based on the results presented in Table 7 and Figure 13). It is important for the authors to clarify that the BAU estimates represent a lower bound. Again, I acknowledge that it is beyond the scope of the paper to incorporate the environmental cost of high chloride concentrations. That said, it would be useful to consider whether some estimates of the environmental cost of chloride concentrations have already been documented in the literature. If such estimates exist, I’d like to see them discussed (briefly) in the manuscript so that the reader can get a sense of the magnitude of the environmental cost associated with the BAU scenario. At minimum, the analysis needs to stress this limitation in order not to make the BAU solution appear as the most cost-effective one.

Response: We appreciate your comment on the higher environmental cost of the BAU scenario, which has not been quantified in the paper. We have added a paragraph on Discussion of Cost Estimates at the end of the Results and Discussion section of the revised manuscript to clarify that the BAU scenario represents a lower bound of total costs including environmental costs. We have also clarified that owing to related conservative assumptions and data limitations, our cost estimates for the CS and RO-EC scenarios are upper bounds of the actual costs of these scenarios. We have also included the following information on estimates of environmental costs of the BAU scenario from the literature, in this sub-section.

The BAU scenario leads to underestimating the total costs to society from continued chloride pollution including potential losses to valuable ecosystem services, i.e. benefits received from nature [1] such as water quality, nutrient cycling, wildlife habitat, recreation, and clean drinking water (which includes the public health costs of excessive chloride in drinking water). There has been little research estimating the environmental costs of chloride pollution. We discuss a few examples from the literature here to provide a perspective on these unaccounted environmental costs of the BAU scenario. Estimates of ecosystem service valuation based on peoples’ willingness to pay (WTP) for restoration and indirect estimates through species like freshwater mussels that are sensitive to chloride pollution [2] and also provide ecosystem services [3], provide useful reference on these costs. For example, a study in Brazil found local residents willing to pay increases to their current water bills by 0.81% to 1.25% of their average income, to restore water quality in the Paraíba do Sul River, which was contaminated with industrial and agricultural pollutants over many years [4]. With regard to freshwater mussels, a study in the Kiamichi River in Southeastern Oklahoma, found three ecosystem services: biofiltration, nutrient cycling, and nutrient storage, declined by 60% over a 20-year period in proportion to the decline in freshwater mussels in the river [5]. A survey of stakeholders for the same river found habitat provision and water quality had the highest economic value with annual WTPs per household at $20.34 and $9.59 [6]. Chloride being toxic to fish [7], valuation estimates for fishing are also indicative of potential losses from chloride pollution. The freshwater fishing industry generates $2.4 billion in expenditures annually in Minnesota [8], and $30 billion [9] nationally, so that excessive chloride contamination of surface water could affect fishing, recreation, and tourism. Chloride also degrades groundwater sources used for drinking water, and the replacement costs of chloride-contaminated drinking water through desalination were found to be substantial, estimated as $3600 per acre foot of drinking water [10]. Groundwater salinity can also result in agricultural losses. For example increasing salinity of irrigation water was associated with declining farmland values in California’s Central Valley [11]. The economic costs of the public health crisis in Flint, Michigan, caused by corrosion in the drinking water distribution system from elevated levels of chloride in source water, have been estimated to be more than $360 million in averting expenditures and government spending [12]. The BAU scenario would also have an added environmental loss from a higher carbon footprint compared to centralized softening, which was found to mitigate climate change by 0.11 Million tonnes CO2 eq./year in a life cycle study of multiple softening options in the Netherlands [13]. 

2) It would be interesting to clarify what share of the CS-RO and CS-Lime scenarios costs would go towards covering home-softener removal (rebate plus labor cost associated with the removal) vs. capital and O&M costs of technology installation at the water treatment plant. Some of that information is provided in the supporting information .csv file that the authors have made available to the reviewers, but it would be useful to include a summary of that information in the main manuscript.

Response: The cost of removing home softeners can depend on several factors including public preferences and political will, which may not be quantifiable. Our analysis accounted for a softener removal cost of $2000 including removal fees for a standard point-of-entry water softener, payable by the user or homeowner decommissioning the removal. We did not incorporate explicit rebates into the softener removal cost as the agency’s rebate grant program is new and does not have sufficient data on applicants. However, we annualized the total payment to $140.76 per household using a 20-year payback period at a 3.5% interest rate accounting for availability of softener removal assistance such as low interest loans from the agency’s Clean Water Partnership loan program [14]. From our results, the median share of the softener removal cost in total annual user cost per household is 5.3% for CS-Lime and 8.4% for CS-RO scenarios. Our analysis provides a conservative (upper bound) estimate of home softener removal costs. We have included a summary of this information in the Methods sub-section of the revised manuscript.

3) Related to point #2 above, the authors mention (on page 17) that funds are available from a rebate grant program towards retiring existing home-softeners. It would be useful to consider several alternatives for the retirement of existing home-softeners. Different alternatives could have different cost-effectiveness implications. For example, would all households in a given city need to retire their softeners? Or would it be more cost-effective to focus on households with high water consumption? If rebates are provided as an incentive to retire a home-softener, how would each household rebate be determined? I assume rebates will be determined based on the NPV of the remaining lifetime of the home-softener, but perhaps there are other ways. An alternative approach (instead of offering a rebate) would be to offer a discount to a household’s water utility bill over the course of a few months/years. That might help spread out the cost of the retirement investment (the personnel cost of removing the water softener would still need to be paid in the present).

Response: In this paper, we did not consider rebates as part of the analysis because the rebate grant program is new and does not yet have data that can be used for this purpose. However, we did consider alternatives to retiring all home softeners and found they would not lead to sufficient chloride reduction to meet chloride water quality standards, since most cities that have reasonable potential (RP) for exceeding their effluent limits, are currently discharging substantially above their limits. A very few cities (less than 5%) would possibly meet limits by retiring a part of their softeners but a policy targeting only a handful of cities would be almost impossible to implement. This reasoning also works for policies targeting households with high water consumption. For this reason, our estimate of the softener removal costs presents an upper bound estimate to the user or home-owner. This also means that despite considering the highest levels of user cost for the centralized softening scenarios, we still obtain the result of centralized softening to be more cost-effective than the alternative of end-of-pipe treatment, and that centralized RO-softening can be similar in costs to the BAU or status-quo option to the users. We do note that, for centralized softening to reach its goal of sufficient reduction in chloride pollution, all home softeners in a given city would have to be retired/removed, and this assumption is incorporated in the analysis.

Rebates will be determined based on the number of applications for home softener removals, or related costs (switching to higher efficiency softeners or optimizing existing softeners) and program funding availability, once centralized softening is approved for many cities. Rebate amounts to individual households and method of issuance are typically determined by the cities. Rebates starting from $100 to $2000 (covering 100% of home softener removal costs) are in operation in several cities and states across the country as discussed in our agency’s new rebate grant program guidelines [15]. They are also typically issued as a credit to the user’s water bill and would be similar to a discount. The Water Infrastructure Fund (WIF) [16], an assistance program administered jointly by the Public Facilities Authority (PFA) and MPCA (our agency) offers long term low interest loans to water utilities for infrastructure projects such as centralized softening and simultaneously issues grants to protect users from higher water costs due to these infrastructure projects, by capping water bill increases to 1.4% of median household income. This assistance to users works similar to a discount offsetting higher water costs over time. 

Minor comment:

1) Page 17: Can you provide some more detail as to why grants increased so much in 2018 (Figure 9)? Is that increased spending likely to continue into the future?

Response: The increase in grants was the result of recent legislative changes that allowed cities access to two infrastructure grant and loan programs: the Point source implementation Grant (PSIG) [16], and the Water Infrastructure Fund (WIF) [17], for drinking water infrastructure projects, for the first time in 2017, and also increased the total amount of potential support from $3 million to $7 million. The statutory changes are documented here for each program [18, 19] 

Prior to these amendments, PSIG and WIF were mainly accessible by wastewater projects. PSIG and WIF are partially funded by Minnesota’s Clean Water Fund (CWF) [20], created in 2009 as part of Minnesota’s Clean Water Land and Legacy Amendment, a constitutional amendment in which the state’s citizens voted to raise their sales taxes by 3/8th of 1 percent to fund a cleaner environment including improved water quality [21].

PSIG and WIF are jointly administered by the Public Facilities Authority (PFA) and our agency (MPCA), and issue grants and low-interest loans to cities annually for water infrastructure projects, based on a variety of factors related to project funding needs including population, median household income (MHI), age and condition of existing infrastructure, and the existence of effluent limits for specific pollutants such as chloride faced by their wastewater treatment plants. The idea behind the amendments was to provide cities with more flexibility in attaining clean water goals by including drinking water infrastructure projects in their application. As centralized softening will have to be installed in drinking water plants to realize the benefits to water quality from reduced chloride pollution as well as provide an alternative way to soften water, these legislations are particularly helpful to cities for corresponding upgrades needed to their drinking water infrastructure. The PSIG program is specifically designed to help communities meet effluent limits on their wastewater systems and contributes up to 80% of the total project costs with a cap of $7 million, while the WIF is a long term low interest loan program, which issues loans to cities for infrastructure projects, while protecting ratepayers by capping their water bills to at 1.2%-1.4% of MHI through grants based on affordability. Consequently access to these programs opened up substantial new funds for communities directed towards needed drinking water infrastructure projects, as shown in Figure 9. This increased funding is likely to continue in the future as the legislations supporting them are permanently in the state statute. We have provided a summary of these details close to Figure 9 in the revised manuscript.

Reviewer #2: The paper is generally well-written (although it could use a proofreading, especially for comma use), it is of limited value to anyone outside of Minnesota as written. Without a more extensive attempt to relate this research to more general water quality problems and solutions that would be applicable in other states or countries, this paper reads very much as a local analysis for local water quality benefit.

Response: We are happy that you find our paper to be well-written. We have reviewed the manuscript and corrected it for grammatical errors and style, including comma use. 

The applicability of our findings have a broad national context and our paper uses Minnesota as a case study. Our paper investigates Minnesota’s chloride pollution problem using data from Minnesota communities and finds centralized softening to be the most environmentally effective and cost effective solution for chloride pollution and its impacts. The findings of this study are applicable nationally because the costs of affordable chloride reduction alternatives for municipalities are generalizable, scalable and allow for chloride reductions everywhere. 

EPA’s nationally recommended 230 mg/L chloride water quality criteria [22] for surface freshwater dates back to 1988, and recent advances in chloride toxicology finds this criteria needs updating to protective of sensitive aquatic species such as mayflies [23, 24] and mussels [25, 2]. Minnesota is among thirty-five states that have surface water quality criteria for chloride that either follow EPA’s national criteria, localized updates to the national criteria or have criteria to protect irrigation or public water supplies. If any state, tribe or territory were to update their aquatic life criteria using the best available science, it is likely that wastewater dischargers would face more restrictive chloride discharge requirements and many dischargers would need to find ways to further reduce chloride in their discharge. Chloride concentrations are increasing in waterbodies across the United States as shown in Figure 2 and home water softening is a contributor to excess chloride in many states including Arizona, California, Connecticut, Illinois, Iowa, Michigan, Ohio, Texas, and Wisconsin. We attempted to quantify chloride loading from water softeners on a regional or national level, but abandoned the effort because it is exceedingly difficult and resulted in unreasonable uncertainty. Based on general observation, areas of the country with hard water are very likely to have a high chloride load from widespread use of water softeners. In areas of hard water, municipal wastewater plants are increasingly being required to reduce chloride loading from water softeners to comply with chloride wastewater discharge requirements. Since there is no national strategy on reducing chloride from water softeners, affected cities have employed a patchwork of approaches ranging from state regulations allowing local water softener bans (California [26], Connecticut [27], Texas [28]) to providing local incentives to install high efficiency softeners (Scottsdale, Arizona [29]; Lake Geneva , Wisconsin [30]) to requiring cities to install centralized softening (Fort Dodge, Iowa [31]) to installing chloride treatment technologies at the wastewater plant (Santa Clarita, California [32]). 

Excessive chloride pollution of surface and groundwater could also lead to increased corrosion of water infrastructure such as drinking water distribution systems; therefore, controlling corrosion is an important part of avoiding public health crises related to unsafe drinking water such as occurred in Washington DC [33] and Flint Michigan [34]. The current increasing trend of chloride in drinking water, including groundwater in several cities, such as Madison, Wisconsin [35], and Chicago, Illinois [36], highlights the need for a solution such as centralized softening, which unlike home water softening, avoids adding chloride to the environment and helps prevent such public and environmental health threats nationwide. We have added a paragraph on the national context of our problem and applicability of results in the Introduction of the revised manuscript.

Reviewer #3: The manuscript “Centralized softening as a solution to chloride pollution: An empirical analysis based on Minnesota cities” by Bakshi et al. is a well written, comprehensive discussion of the potential for centralized water softening as a solution for a challenging problem, chloride pollution. I think the article is in good shape and a useful contribution to the literature (I am less familiar with the engineering side, and it may be well discussed there, but this topic would benefit the discussion on the env. sci. side. I have some suggestions for improvement, listed below.

Figures in general: The figures should be more focused. For example, the national USGS map (Figure 2) could be cited and a map focused on MN/the 5-6 state region replace that map. Use of those maps doesn’t necessarily add much to the paper.

Response: We have another reviewer ask about the scope of our paper beyond Minnesota and we decided to add a paragraph in the Introduction of the revised manuscript, to address this question, stating that our problem and results have a national scope and justifying it with examples from other states. In this context, Figure 2, which shows chloride concentrations have an increasing trend in multiple parts of the United States, is relevant as evidence. We do have Figure 3 (a higher resolution version will be uploaded to replace the existing Figure 3) for showing the chloride problem in Minnesota in more detail. 

ll 77: Probably better to call “bugs” “insects” or “macroinvertebrates” or the like

Response: We have changed “bugs” to “insects” in the revised manuscript.

ll 99: It would be useful to also report this as a proportion of the population (i.e., not just a proportion of the counts)

Response: We have modified the sentence to add the population proportion in the revised manuscript.

ll 113: this is an incomplete sentence

Response: We have fixed this sentence in the revised manuscript so it does not read like an incomplete sentence using a dash and comma and have removed the colon from the original sentence.

ll 125-126: While I am not necessarily advocating you do this in this manuscript, it would be helpful to include carbon life cycle implications of these scenarios as you continue this sort of analysis.

Response: We hope to examine and include carbon life cycle implications as possible future research extensions of this manuscript. According to a life cycle assessment of multiple softening options including home-based ion-exchange softening, centralized softening had the lowest carbon footprint, and in addition, was found to reduce climate change (had a net negative carbon footprint of -0.11 Million tonnes CO2 eq./year) when its beneficial effects on household appliances were included [13]. We cited this research as an additional benefit of centralized softening over home-based softening, in the revised manuscript.

ll 153: make it clear this is 75% of the residents (just to be scrupulous)

Response: We have changed it to 75% of residents. 

Table 1; Data sources should be cited in the caption so this can stand alone.

Response: Data sources for Table 1 are now cited in the caption in the revised manuscript.

Table 2: It’s not clear what the green highlighting means. Can you clarify in the caption?

Response: The green highlighting indicates options that meet criteria in columns 5, 6, and 7 (where 7 is implementation feasibility based on permitting) and hence considered in analysis. We have now added this content to the caption of Table 2.

Table 3: Consider spelling out RP in the caption to help the reader.

Response: We have spelled out RP in the caption to Table 3 in the revised manuscript.

Page 21, “Data” section: When these data are pulled from publicly available databases, citations really should be provided.

Response: We have added citations for the publicly available data in the Data section.

References: I strongly encourage the use of more original/authoritative sources than Wikipedia. I agree, it’s a tremendous resource, but, for example the grains per gallon is stripped directly from other fundamental information sources such as the “Standard Methods for Analysis of Water and Wastewater.” Citations help those important community efforts. Don’t starve ‘em

Response: We have replaced the Wikipedia sources with other sources in the references.

Related question:

Do changes in geochem (e.g., a more sulfate dominated mix, in places like the Red River Valley) impact these cost estimates? I.e., does the sulfate foul things?

Response: Based on engineering findings, changes in geochemical composition, for example a more sulfate dominated mix, is unlikely to change these cost estimates in any significant way.

Sulfate at the typical concentrations found in Minnesota does not cause fouling in drinking water systems. The highest concentrations of sulfate in drinking water sources is found in Southwest Minnesota and can be as high as 400 mg/L sulfate in cities such as Marshall and Lakefield. Both Lakefield and Marshall, use centralized softening to soften their ‘high’ sulfate water without significant problems. All things being equal, it is easier to operate and design a centralized softening system when sulfate concentrations are ‘low’, but the costs of treating a ‘high’ sulfate water are typically not significantly different than treating a ‘low’ sulfate water. 

References

1. Reid WV, Mooney HA, Cropper A, Capistrano D, Carpenter SR, Chopra K, Dasgupta P, Dietz T, Duraiappah AK, Hassan R, Kasperson R. Ecosystems and human well-being-Synthesis: A report of the Millennium Ecosystem Assessment. Island Press; 2005.

2. Wang N, Kunz JL, Dorman RA, Ingersoll CG, Steevens JA, Hammer EJ, Bauer CR. Evaluation of chronic toxicity of sodium chloride or potassium chloride to a unionid mussel (Lampsilis siliquoidea) in water exposures using standard and refined toxicity testing methods. Environmental toxicology and chemistry. 2018 Dec;37(12):3050-62.

3. Vaughn CC. Ecosystem services provided by freshwater mussels. Hydrobiologia. 2018 Mar 1;810(1):15-27.

4. Vásquez WF, de Rezende CE. Willingness to pay for the restoration of the Paraíba do Sul River: A contingent valuation study from Brazil. Ecohydrology & Hydrobiology. 2019 Oct 1;19(4):610-9.

5. Vaughn CC, Atkinson CL, Julian JP. Drought‐induced changes in flow regimes lead to long‐term losses in mussel‐provided ecosystem services. Ecology and Evolution. 2015 Mar;5(6):1291-305.

6. Castro AJ, Vaughn CC, García-Llorente M, Julian JP, Atkinson CL. Willingness to pay for ecosystem services among stakeholder groups in a South-Central US watershed with regional conflict. Journal of Water Resources Planning and Management. 2016 Sep 1;142(9):05016006.

7. Corsi SR, Graczyk DJ, Geis SW, Booth NL, Richards KD. A fresh look at road salt: aquatic toxicity and water-quality impacts on local, regional, and national scales. Environmental science & technology. 2010 Oct 1;44(19):7376-82

8. U.S. Department of the Interior, U.S. Fish and Wildlife Service, and U.S. Department of Commerce, U.S. Census Bureau. 2011 National Survey of Fishing, Hunting, and Wildlife-Associated Recreation. Minnesota. 2013. Available from: https://www.census.gov/prod/2013pubs/fhw11-mn.pdf

9. US Fish and Wildlife Service. US Department of Commerce and US Census Bureau. 2018. 2016 National Survey of Fishing, Hunting, and Wildlife-Associated Recreation. US Fish and Wildlife Service, US Department of Commerce and US Census Bureau. Available from: https://www.fws.gov/wsfrprograms/subpages/nationalsurvey/nat_survey2016.pdf

10. California Public Utilities Commission. What Will Be the Cost of Future Sources of Water for California? 2016. Available from: https://www.cpuc.ca.gov/uploadedFiles/CPUC_Public_Website/Content/About_Us/Organization/Divisions/Policy_and_Planning/PPD_Work/PPD_Work_Products_(2014_forward)/PPD%20-%20Production%20costs%20for%20new%20water.pdf

11. Mukherjee M, Schwabe KA. Where's the salt? A spatial hedonic analysis of the value of groundwater to irrigated agriculture. Agricultural Water Management. 2014 Nov 1;145:110-22.

12. Christensen P, Keiser D, Lade G. Economic effects of environmental crises: Evidence from Flint, Michigan. Michigan (June 20, 2019). 2019 Jun 20.

13. Beeftink M, Hofs B, Kramer O, Odegard I, van der Wal A. Carbon footprint of drinking water softening as determined by life cycle assessment. Journal of Cleaner Production. 2020 Jul 22;278:123925.

14. Minnesota Pollution Control Agency. Clean Water Partnership Loan Program. Available from: https://www.pca.state.mn.us/water/cwp-loans

15. Minnesota Pollution Control Agency. Guide to developing a local water softener rebate program; 2019. Available from: https://www.pca.state.mn.us/sites/default/files/p-tr1-53.pdf

16. Minnesota Department of Employment and Economic Development. Public Facilities Authority. Infrastructure Funds & Programs. Point Source Implementation Grant Program. Point Source Implementation Grants. 2020. Available from: https://mn.gov/deed/pfa/funds-programs/point-source-grants.jsp

17. Minnesota Department of Employment and Economic Development. Public Facilities Authority. Infrastructure Funds & Programs. Water Infrastructure Fund. Available from: https://mn.gov/deed/pfa/funds-programs/wastewater.jsp

18. Minnesota Legislature. Office of the Revisor of Statutes. 2017 Minnesota Statutes. 446A. 073 Point Source Implementation Grants. 2017. Available from: https://www.revisor.mn.gov/statutes/2017/cite/446A.073

19. Minnesota Legislature. Office of the Revisor of Statutes. 2017 Minnesota Statutes. 446A. 072 Water Infrastructure Funding Program. 2017. Available from: https://www.revisor.mn.gov/statutes/2017/cite/446A.072

20. Minnesota Legislature. Office of the Revisor of Statutes. 2009 Minnesota Statutes. 114D.50 Clean Water Fund. 2009. Available from: https://www.revisor.mn.gov/statutes/2009/cite/114D.50

21. Minnesota’s Legacy. About the Funds. Legacy Funds. Clean Water Land and Legacy Amendment. 2017. Available from: https://www.legacy.mn.gov/about-funds

22. Environmental Protection Agency. Ambient Water Quality Criteria for Chloride – 1988. EPA 440/5-88-001. February 1988.

23. Soucek DJ, Dickinson A. Full‐life chronic toxicity of sodium salts to the mayfly Neocloeon triangulifer in tests with laboratory cultured food. Environmental Toxicology and Chemistry. 2015 Sep;34(9):2126-37.

24. Kefford BJ. Why are mayflies (Ephemeroptera) lost following small increases in salinity? Three conceptual osmophysiological hypotheses. Philosophical Transactions of the Royal Society B. 2019 Jan 21;374(1764):20180021.

25. Prosser RS, Rochfort Q, McInnis R, Exall K, Gillis PL. Assessing the toxicity and risk of salt-impacted winter road runoff to the early life stages of freshwater mussels in the Canadian province of Ontario. Environmental Pollution. 2017 Nov 1;230:589-97.

26. California Legislative Information. Health And Safety Code. Division 103. Part 12. Drinking water. Chapter 5. Water Equipment and Control [116775-116890]. Available from: https://leginfo.legislature.ca.gov/faces/codes_displayText.xhtml?lawCode=HSC&division=104.&title=&part=12.&chapter=5.&article=1.

27. Connecticut Bureau of Materials Management and Compliance Assurance. Conn. Gen. Stat. § 22a-430(g). 2018. Available from: https://www.epa.gov/sites/production/files/2018-01/documents/draftctgpdlfwtww.pdf

28. Texas Administrative Code. Title 30 part 1 Chapter 285. Subchaper D Rule 285.37. 2004. Available from: https://texreg.sos.state.tx.us/public/readtac$ext.TacPage?sl=R&app=9&p_dir=&p_rloc=&p_tloc=&p_ploc=&pg=1&p_tac=&ti=30&pt=1&ch=285&rl=37

29. City of Scottsdale. Water. Rebates. 2020. Available from: https://www.scottsdaleaz.gov/water/rebates

30. Lake Geneva Utility Commission. Wastewater Utility. Water Softener Rebate Program. 2017. Available from: https://www.lgutilitycommission.com/wastewaterutility

31. Shea, Bill. Fort Dodge to tackle hard water woes. The Messenger. 2017 April 25. Available from: https://www.messengernews.net/news/local-news/2017/04/fort-dodge-to-tackle-hard-water-woes/

32. City of Santa Clarita. Environmental Services. State-Mandated Chloride (Salt) Limits in the Santa Clarita Valley. 2003. Available from: https://www.santa-clarita.com/city-hall/departments/neighborhood-services/environmental-services/chloride-salt

33. Environmental Protection Agency. Lead in DC Drinking Water. Corrosion Control Research. 2016. Available from: https://archive.epa.gov/region03/dclead/web/html/corrosion_research.html

34. Natural Resources Defense Council. Flint Water Crisis. 2020. Available from: https://www.nrdc.org/flint

35. City of Madison. Madison Water Utility. Road Salt Study at Well 14. 2020. Available from: https://www.cityofmadison.com/water/projects/road-salt-study-at-well-14

36. Metropolitan Water Reclamation District of Greater Chicago. Public Affairs. Issues and Initiatives. Chicago Area Waterways Chloride Initiative Work Group. 2018. Available from: https://legacy.mwrd.org/irj/portal/anonymous?NavigationTarget=navurl://11b3b3c22a74fe92b74ec8364bb8f264

---

## [Decision Letter · Decision Letter 1]

4 Jan 2021

PONE-D-20-25024R1

Centralized softening as a solution to chloride pollution: An empirical analysis based on Minnesota cities

PLOS ONE

Dear Dr. Bakshi,

Thank you for submitting your manuscript to PLOS ONE. After careful consideration, we feel that it has merit but does not fully meet PLOS ONE’s publication criteria as it currently stands. Therefore, we invite you to submit a revised version of the manuscript that addresses the points raised during the review process.

We look forward to receiving your revised manuscript.

Kind regards,

Huan Li

Academic Editor

PLOS ONE

Reviewers' comments:

Reviewer's Responses to Questions

**Comments to the Author**

1. If the authors have adequately addressed your comments raised in a previous round of review and you feel that this manuscript is now acceptable for publication, you may indicate that here to bypass the “Comments to the Author” section, enter your conflict of interest statement in the “Confidential to Editor” section, and submit your "Accept" recommendation.

Reviewer #1: All comments have been addressed

Reviewer #2: (No Response)

Reviewer #3: All comments have been addressed

2. Is the manuscript technically sound, and do the data support the conclusions?

Reviewer #1: Yes

Reviewer #2: Yes

Reviewer #3: Yes

3. Has the statistical analysis been performed appropriately and rigorously? 

Reviewer #1: Yes

Reviewer #2: I Don't Know

Reviewer #3: Yes

4. Have the authors made all data underlying the findings in their manuscript fully available?

Reviewer #1: Yes

Reviewer #2: Yes

Reviewer #3: Yes

5. Is the manuscript presented in an intelligible fashion and written in standard English?

Reviewer #1: Yes

Reviewer #2: Yes

Reviewer #3: Yes

6. Review Comments to the Author

Reviewer #1: (No Response)

Reviewer #2: The discussion on pages 6-7 adequately addresses my concern about the national applicability of this case study, but it should come earlier in the manuscript introduction and be associated with Figure 2--i.e., the paper should move from a national-level discussion of the problem to the Minnesota case study so that the case study is adequately framed from the beginning. The current back-and-forth between Minnesota and national considerations in the Introduction is jarring. It also might be beneficial to circle back to the national implications at the end of the paper.

Reviewer #3: This is ready. A couple things caught when re-reading:

In the abstract lime shouldn't be capitalized

ll76 I think you mean reduction of Cl

pg 17, last paragraph. To say the hardness is the same is too definite. Similar would be better (e.g., lawn irrigation would make the output higher.

7. PLOS authors have the option to publish the peer review history of their article (what does this mean?). If published, this will include your full peer review and any attached files.

Reviewer #1: **Yes: **Nikolaos Zirogiannis

Reviewer #2: No

Reviewer #3: No

---

## [Author Response · Author response to Decision Letter 1]

22 Jan 2021

Response to Reviewers for manuscript titled “Centralized softening as a solution to chloride pollution: An empirical analysis based on Minnesota cities

January 22, 2021

Dear editor and reviewers, 

Thank you for reviewing our manuscript and for your helpful comments. This rebuttal letter contains our responses to each of the comments from the reviewers (part 6 of the letter to authors from Plos-One) for the second revision of the manuscript. Please let us know if you have any question/s.

Sincerely,

Baishali Bakshi (on behalf of the author team)

6. Review Comments to the Author

Reviewer #1: (No Response)

Reviewer #2: The discussion on pages 6-7 adequately addresses my concern about the national applicability of this case study, but it should come earlier in the manuscript introduction and be associated with Figure 2--i.e., the paper should move from a national-level discussion of the problem to the Minnesota case study so that the case study is adequately framed from the beginning. The current back-and-forth between Minnesota and national considerations in the Introduction is jarring. It also might be beneficial to circle back to the national implications at the end of the paper.

Response: We agree and have reorganized the content of the introduction so that the national context discussion now precedes and leads to the discussion on the Minnesota case study. This reorganization has led to changes in the position of Figures 1 and 2: now Figure 2 is actually Figure 1 and vice versa. We have also updated the numbering of the in-text citations and synced them in the references. Finally, we have added a sentence in the conclusion that circles back to the national implications of the problem discussed in the introduction.

Reviewer #3: This is ready. A couple things caught when re-reading:

In the abstract lime shouldn't be capitalized

ll76 I think you mean reduction of Cl

pg 17, last paragraph. To say the hardness is the same is too definite. Similar would be better (e.g., lawn irrigation would make the output higher.

Response: Thank you. We have made the following changes based on your comments. 

a. We have un-capitalized lime in the abstract. 

b. Here is the sentence from lines 76-77: ‘‘Chloride in WWTP discharges is generally viewed as infeasible to treat due to high cost and limited availability of treatment technologies”.

We prefer the term ‘remove’ to replace the term ‘treat’ because ‘removing’ is a more specific term from the perspective of an engineer. Removing means taking chloride out of the wastewater. The result of “removing” is a reduction in chloride but the converse isn’t true. Based on your comment and the above, we have revised the sentence as:

“Removing chloride from WWTP discharges is infeasible due to high cost and limited availability of treatment technologies”, in the revised manuscript.

c. We have replaced ‘the same’ with ‘similar’ in the last paragraph of page 17 in the revised manuscript.

Note: Apart from these changes, we have made some additional changes to correct typos, add more references, and add more updated/relevant content. These additional changes are listed below. 

1. We un-capitalized ‘Lime’ in Lime-softening throughout the manuscript.

2. We updated reference #16 as the corresponding manuscript has just been published. 

3. We corrected a typo (crossed out Effluent Limits Unit) from the affiliation of the corresponding author.

4. We added the word ‘sources’ to sentence 2 (line 29) in the abstract.

5. We added the word ‘statewide’ to line 389, page 16 in the Revised Manuscript with Track Changes.

6. We explained corrosion control and operator costs as part of O&M costs of CS technologies, on page 24 and referred to these in the ‘Factors considered’ sub-section on page 33 of the Revised Manuscript with Track Changes.

7. We added 6 additional references, as required by the revision and those excluded by oversight earlier.

8. We acknowledged our sister agency the Minnesota Department of Health in the acknowledgment section, which we had omitted before owing to oversight.

---

## [Editor Report · Decision Letter 2]

25 Jan 2021

Centralized softening as a solution to chloride pollution: An empirical analysis based on Minnesota cities

PONE-D-20-25024R2

Dear Dr. Bakshi,

We’re pleased to inform you that your manuscript has been judged scientifically suitable for publication and will be formally accepted for publication once it meets all outstanding technical requirements.

Kind regards,

Huan Li

Academic Editor

PLOS ONE
---

## [Editor Report · Acceptance letter]

27 Jan 2021

PONE-D-20-25024R2 

Centralized softening as a solution to chloride pollution: An empirical analysis based on Minnesota cities 

Dear Dr. Bakshi:

I'm pleased to inform you that your manuscript has been deemed suitable for publication in PLOS ONE. Congratulations! Your manuscript is now with our production department. 

Kind regards, 

on behalf of

Dr. Huan Li 

Academic Editor

PLOS ONE